# A TRIANGLE Enables Multimodal Alignment Beyond Cosine Similarity

**Giordano Cicchetti, Eleonora Grassucci, Danilo Comminiello**
Department of Information Engineering, Electronics, and Telecommunications
Sapienza University of Rome, Italy
`{name.surname}@uniroma1.it`

## Abstract

Multimodal learning plays a pivotal role in advancing artificial intelligence systems by incorporating information from multiple modalities to build a more comprehensive representation. Despite its importance, current state-of-the-art models still suffer from severe limitations that prevent the successful development of a fully multimodal model. Such methods may not provide indicators that all the involved modalities are effectively aligned. As a result, some modalities may not be aligned, undermining the effectiveness of the model in downstream tasks where multiple modalities should provide additional information that the model fails to exploit. In this paper, we present TRIANGLE: TRI-modAl Neural Geometric LEarning, the novel proposed similarity measure that is directly computed in the higher-dimensional space spanned by the modality embeddings. TRIANGLE improves the joint alignment of three modalities via a triangle-area similarity, avoiding additional fusion layers or pairwise similarities. When incorporated in contrastive losses replacing cosine similarity, TRIANGLE significantly boosts the performance of multimodal modeling, while yielding interpretable alignment rationales. Extensive evaluation in three-modal tasks such as video-text and audio-text retrieval or audio-video classification, demonstrates that TRIANGLE achieves state-of-the-art results across different datasets improving the performance of cosine-based methods up to 9 points of Recall@1. Code and checkpoints available at `https://github.com/ispamm/TRIANGLE/`.

## 1 Introduction

Humans perceive reality as a mixture of signals coming from different modalities registered through diverse senses, such as sounds perceived by the ears or vision by the eyes, and process these multimodal inputs to understand the scene. In the last few years, foundational models have attempted to emulate this understanding system by aligning pairs of modalities using contrastive learning approaches. The first valuable contribution in this sense, CLIP Radford et al. (2021), aligns image and textual latent representation with a contrastive loss based on the cosine similarity between the two vectors. CLIP established the *de-facto* receipt for aligning multimodal latent features and the subsequent methods, such as CLAP for audio-text alignment Elizalde et al. (2023), have followed the same scheme.

Later, several works focused on extending the CLIP approach to more than two modalities, incorporating audio, depth, or other modalities to build a more representative embedding space Girdhar et al. (2023); Zhu et al. (2024); Ruan et al. (2023); Wang et al. (2024). All these methods still rely on the pairwise cosine similarity between modalities. Specifically, they select an anchor modality, which may be image Girdhar et al. (2023), audio Ruan et al. (2023), or language Zhu et al. (2024), and then align the embeddings of all the other modalities one-by-one to the anchor embedding. However,

39th Conference on Neural Information Processing Systems (NeurIPS 2025).

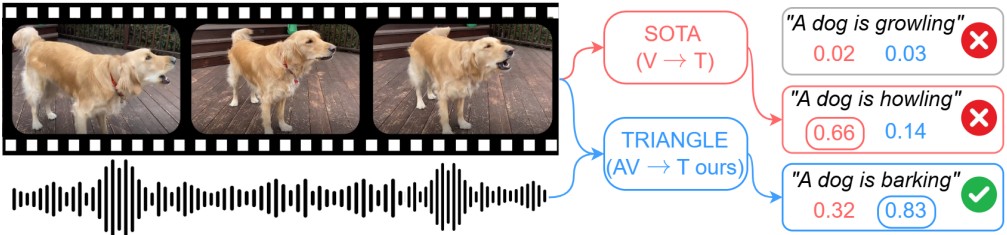

Figure 1: Current state-of-the-art (SOTA) methods struggle to incorporate the third audio modality and fail in video-to-text tasks, incorrectly assigning captions such as "*A dog is howling*", as the frames may be similar. On the contrary, the proposed TRIANGLE method effectively leverages all modalities together, bringing crucial information to discriminate among the captions, and retrieving the correct caption "*A dog is barking*".

this alignment only ensures that all modalities are aligned with the anchor, without providing any indicators for the alignment between non-anchor modalities. Therefore, if the video is aligned with the language as the anchor modality, and the audio is also aligned with the language, there are no geometric indicators that the audio and video modalities are aligned with each other. Such a limitation may undermine the effective applicability of these models in real-world multimodal scenarios. Indeed, they mostly perform experiments on two-modal tasks where they just test the alignment between one modality and the anchor one, as their pairwise similarity loss does not allow them to test three modalities at once. However, even in these scenarios, such models often fail when sample data includes a third modality that may be crucial to accurately discriminate and truly understand the content Yoon et al. (2023). As an example, in widespread video-text retrieval tasks, conventional state-of-the-art (SOTA) models focus on video frames to solve the task, while the audio information is instead often crucial to properly retrieve the correct caption. Figure 1 shows an example of this task in which the SOTA model fails as it is not able to leverage the audio information, while the proposed one exploits both the modalities and correctly retrieves the caption. To partially overcome the limitation due to the use of only two modalities, other works have proposed to fuse multiple modalities via MLP layers Chen et al. (2023b), supplementary loss functions Wang et al. (2024); Chen et al. (2023b) or additional architectural strategies Li et al. (2021). Very recently, GRAM Cicchetti et al. (2025) and Symile Saporta et al. (2024) proposed general alternative similarity objectives that scale to an arbitrary number $n$ of modalities by minimizing, respectively, the parallelotope volume and a total-correlation bound.

Unfortunately, while these methods obtain improved alignment performance, some of them still lack geometrical indicators and fail to provide insights on how each modality impacts the final outcome, thereby reducing the interpretability of the results as well. Additionally, the majority of the recent works come up with their own datasets Zhu et al. (2024); Chen et al. (2023b,a); Saporta et al. (2024), placing a strong emphasis on data quality and on scaling up neural models to billions of parameters Wang et al. (2024), yet still relying on the same similarity measure.

In this paper, we present TRIANGLE: TRI-modAl Neural Geometric LEarning, a novel method capable of addressing the aforementioned limitations. The proposed method directly works within the space spanned by all embedding vectors, aligning them altogether. In this space, the extremities of the vectors form the vertices of a 2D polygon, whose shape and area show interesting insights into the semantic affinity among modalities. In the most common case of video, audio, and language modalities, such a polygon is a triangle, whose area is strictly related to the alignment of all the modalities. Geometrically, the smaller the area of the triangle, the closer (i.e., the more aligned) the vectors of the three modalities are to each other. Conversely, a larger area indicates that the vectors are more orthogonal, suggesting they point in opposite directions and probably represent misaligned data. Therefore, TRIANGLE proposes using triangle area minimization as the similarity metric to align three-modal vectors in their higher-dimensional space, without relying on pairwise comparisons. This method effectively integrates the information from the three modalities, as shown in Fig. 1, where TRIANGLE exploits both the video and audio information to retrieve the correct caption. TRIANGLE sets a new state of the art in video-text retrieval across datasets such as MSR-VTT, DiDeMo, ActivityNet, and VATEX, as well as in audio-text retrieval and classification in AudioCaps and VGGSound, all without requiring new datasets or additional layers, underscoring its superiority

over conventional methods relying on cosine similarities between pairs of modalities and over generic methods designed for joint alignment.

In summary, our contributions are:

1. TRIANGLE, a three-modal alignment method is introduced. TRIANGLE encourages alignment among three modalities, leveraging all the modalities together to perform downstream tasks.

2. TRIANGLE offers an easily interpretable measure of alignment of three modalities, a feature lacking in previous models.

3. TRIANGLE establishes new state-of-the-art results in video-text and audio-text retrieval across diverse datasets and scenarios.

## 2 Related Work

**Two-modal Alignment.** In 2021, CLIP Radford et al. (2021) and ALIGN Jia et al. (2021) paved the way for foundational models capable of aligning two modalities, specifically focusing on images and text. From that framework, several works followed improving the performance and the alignment Uesaka et al. (2024); Ilharco et al. (2021); Zhai et al. (2023); Grassucci et al. (2025). CLIP has also served as the backbone for extending such alignment capabilities to different modalities such as audio and text as in CLAP Elizalde et al. (2023), video and text in CLIP4Clip Luo et al. (2021), point clouds and text in PointCLIP Zhang et al. (2021). Each of these models is based on contrastive learning strategies, which bring similar vectors closer together while pushing dissimilar embeddings further apart. They all rely on the pairwise cosine similarity in the contrastive loss.

**Three-modal Alignment.** More recently, several attempts have been made to better capture the complexity of the reality around us, which usually involves more than two modalities. CLIP4VLA Ruan et al. (2023) proposed to incorporate the audio modality in the CLIP framework, aligning video, audio, and text modalities in pairs using the audio embedding as anchor. Lately, ImageBind Girdhar et al. (2023) introduced a multimodal pre-trained framework that includes multiple modalities such as depth and infrared, considering the image modality as a bridge to all others. Building on this approach, LanguageBind Zhu et al. (2024) showed that using the text modality as the anchor is more effective than using the image modality. Concurrently, different approaches for building large three-modal pretrained embedding models have emerged, including VALOR Chen et al. (2023a), VAST Chen et al. (2023b), mPLUG-2 Xu et al. (2023), and InternVideo2 Wang et al. (2024), all collecting large pretraining datasets and bringing architectural improvements to enhance model performance. However, none of these methods work on the higher-dimensional space spanned by the multimodal vectors, as they primarily rely on the cosine similarity computed on the 2D plane defined by two modalities, or on architectural fusion strategies for multiple modalities. In practice, these approaches are unable to fully leverage the effective multimodal information needed to solve downstream tasks comprehensively. More recently, GRAM Cicchetti et al. (2025) and Symile Saporta et al. (2024) proposed two loss functions that aim to better capture the overall alignment of $n$ modalities. However, despite their generalizability, we will show that they underperform in three-modal (video-audio-text) tasks, showing that an objective tailored to triplets can exploit modality-specific features more effectively than general losses.

## 3 TRIANGLE: TRI-modAl Neural Geometric LEarning

### 3.1 Problem Formulation

Multimodal representation learning aims to derive latent representations from co-occurrent input data modalities. The $i$-th modality is encoded in a latent representation in an $n$-dimensional space using an encoding function $e_i : M_i \to \mathcal{Z}$, with $\mathcal{Z} \in \mathbb{R}^n$. In the case of video-audio-text representation we have a tri-modal representation with three encoding functions: $e_V : V \to \mathcal{Z}$ visual encoder, $e_A : Au \to \mathcal{Z}$ audio encoder, $e_T : T \to \mathcal{Z}$ text encoder.

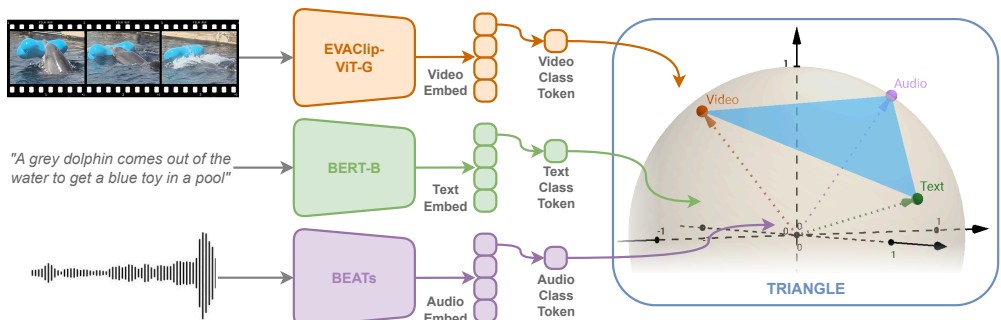

Figure 2: TRIANGLE builds the high-dimensional space spanned by the embeddings generated by the encoders. The embedding vectors of the three modalities lie within a unit hypersphere, where they form a triangle. The area of this triangle is an unambiguous measure of their alignment.

Conventionally, the similarity between two modalities $\mathbf{M}_i, \mathbf{M}_j$ is obtained by computing the cosine of the angle $\theta_{ij}$ between them:

$$\cos(\theta_{ij}) = \frac{\langle \mathbf{M}_i, \mathbf{M}_j \rangle}{||\mathbf{M}_i|| \cdot ||\mathbf{M}_j||} \tag{1}$$

where $\langle \mathbf{M}_i, \mathbf{M}_j \rangle$ is the dot product between modality $\mathbf{M}_i$ and modality $\mathbf{M}_j$, and $||\mathbf{M}_i||$ is the norm.

Intuitively, the closer the cosine value is to $1$, the closer the two vectors are in the hyperdimensional space, meaning that the two embeddings represent similar original content. CLIP Radford et al. (2021) introduces this intuition in the multimodal contrastive loss function, which, given the textual representation $\mathbf{t}$ and the image one $\mathbf{i}$, the number of elements $B$ and the temperature $\tau$ takes the form:

$$\mathcal{L}_{I2T} = -\frac{1}{B} \sum_{i=1}^{B} \log \frac{\exp(\mathbf{i}_i^\top \mathbf{t}_i / \tau)}{\sum_{j=1}^{B} \exp(\mathbf{i}_i^\top \mathbf{t}_j / \tau)} \tag{2}$$

$$\mathcal{L}_{T2I} = -\frac{1}{B} \sum_{i=1}^{B} \log \frac{\exp(\mathbf{t}_i^\top \mathbf{i}_i / \tau)}{\sum_{j=1}^{B} \exp(\mathbf{t}_i^\top \mathbf{i}_j / \tau)}. \tag{3}$$

Whenever the input includes a third modality that needs to be considered by the model, they cannot be naturally compared using cosine similarity. Indeed, cosine similarity is not defined for higher-dimensional spaces and instead, it applies a projection into 2D space. This issue occurs in both the training and inference phases. During training, the most common solution to circumvent this problem is the anchor selection Girdhar et al. (2023); Zhu et al. (2024): one modality is chosen as the anchor and the other modalities are pairwisely aligned to the anchor. This approach guarantees alignment between any modality $\mathbf{M}_i$ and the anchor. However, nothing can be inferred about the alignment between the other modalities. During inference, the problem becomes even worse since conventional methods lack a direct mechanism to compute similarity among three embedding vectors. Therefore, they are forced to rely on only two modalities or develop suboptimal neural fusing strategies. For instance, LanguageBind Zhu et al. (2024) attempts to linearly combine two modalities and then compute similarity with the third one, while methods like UMT Liu et al. (2022), m-PLUG2 Xu et al. (2023) and VAST Chen et al. (2023b) introduce layers that fuse two or more modality embeddings before computing cosine similarity with the remaining one.

Although these suboptimal multimodal strategies show slight improvements in metrics, they fail to fully and effectively exploit the information from the third modality, especially during inference and in downstream tasks, thereby limiting the practical utility of these methods in real-world applications.

### 3.2 The TRIANGLE Solution

We aim at aligning three modalities directly in their natural higher-dimensional space without relying on 2D projections and exploiting the contribution from all the modalities together. To this end, we

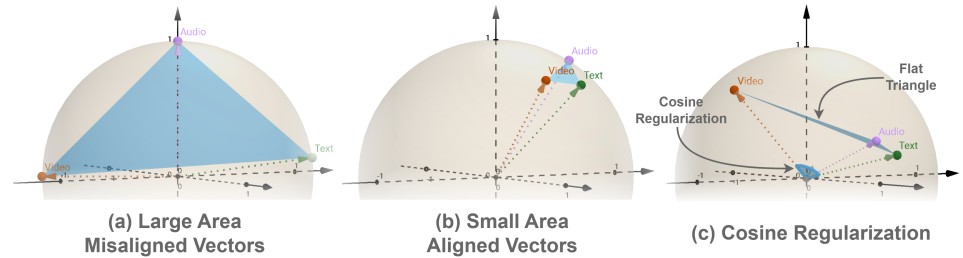

| (a) Large Area Misaligned Vectors | (b) Small Area Aligned Vectors | (c) Cosine Regularization |
|---|---|---|

Figure 3: (a-b)TRIANGLE area as a measure of similarity. Misaligned vectors, large area (a); Aligned vectors, small area (b). (c) In the case of a flat triangle, we apply the cosine regularization, bringing the alignment information between the two modalities on which the downstream task is about. In the example of video-text retrieval, we regularize with the cosine between video and text.

introduce TRIANGLE: TRI-modAl Neural Geometric LEarning, which builds the higher-dimensional space spanned by the modality vectors and computes a novel similarity measure that can be readily applied in downstream tasks.

**Idea.** A generic embedding vector can be interpreted as a point in a multidimensional space $\mathbb{R}^n$, where $n$ is the embedding dimension. In the case of three modalities embedding vectors with unitary norm, the resulting points lie in the unit hypersphere and unambiguously determine a hyperplane (as long as those points are not aligned on the same line). The three modalities embedding vectors draw a triangle, which lies on the aforementioned hyperplane, as the plot in Fig. 2 shows. The area of the triangle spanned by those vectors determines a measure of the similarity of such embeddings and its computation is easy in $\mathbb{R}^n$ as it relies on three dot products.

**Definition 3.1** (**The TRIANGLE area is a measure of similarity**). The area $A$ of a triangle in $\mathbb{R}^n$ is given by:

$$A = \frac{1}{2}\sqrt{\langle \mathbf{u}, \mathbf{u} \rangle \langle \mathbf{v}, \mathbf{v} \rangle - \langle \mathbf{u}, \mathbf{v} \rangle^2}, \tag{4}$$

where $\langle \cdot, \cdot \rangle$ is the dot product and $\mathbf{u} = \mathbf{x} - \mathbf{y}$, $\mathbf{v} = \mathbf{x} - \mathbf{z}$ are two triangle sides computed among the three embeddings $\mathbf{x}$, $\mathbf{y}$, and $\mathbf{z}$ of the three modalities. Proof in Appendix. Intuitively, the smaller the area, the closer the three vectors are, meaning they are well-aligned. Conversely, the largest area occurs when two vectors point in opposite directions, with a corresponding cosine similarity equal to $-1$, while the third vector is orthogonal to them, resulting in an isosceles triangle, as shown in Fig. 3. Therefore, the area of the triangle serves as a direct measure of the similarity among all three vectors, eliminating the curse for pairwise computations.

### 3.3 TRIANGLE Constrastive Loss

In downstream tasks, common models such as ImageBind Girdhar et al. (2023), LanguageBind Zhu et al. (2024), or VAST Chen et al. (2023b) rely on the cosine similarity between the two modalities involved in the task. For example, in the widely used task of video-text retrieval, these models compute the cosine similarity between the embeddings of text and video frames, thus assigning the caption to the video with the highest similarity. In such a common solution, including an additional modality, that could significantly boost the model performance (e.g., the audio in the aforementioned video-text retrieval task), is challenging, as there is no higher-dimensional shared space in which all modalities can be aligned.

We propose to address this widespread limitation by leveraging the TRIANGLE formulation in (9) in conventional contrastive losses, replacing the common pairwise cosine similarity. The proposed brand-new TRIANGLE contrastive losses follow:

$$\mathcal{L}_{D2T} = -\frac{1}{B}\sum_{i=1}^{B}\log\frac{\exp(-A(\mathbf{t}_i, \mathbf{v}_i, \mathbf{a}_i)/\tau)}{\sum_{j=1}^{B}\exp(-A(\mathbf{t}_j, \mathbf{v}_i, \mathbf{a}_i)/\tau)}, \tag{5}$$

$$\mathcal{L}_{T2D} = -\frac{1}{B} \sum_{i=1}^{B} \log \frac{\exp(-A(\mathbf{t}_i, \mathbf{v}_i, \mathbf{a}_i)/\tau)}{\sum_{j=1}^{B} \exp(-A(\mathbf{t}_i, \mathbf{v}_j \mathbf{a}_j)/\tau)}. \tag{6}$$

By exploiting the triangle area $A$ in the contrastive loss, we can effectively integrate information from all modalities and assign the caption to the video according to the minimum area formed by the vector modalities, ensuring a more holistic and accurate alignment. We show how successfully leveraging the three modalities composing the data significantly brings improved performance.

In addition, to further guide the training process and following Chen et al. (2023b), we also employ a data text matching (DTM) loss:

$$\mathcal{L}_{DTM} = \mathbb{E}_{(\mathbf{t},\mathbf{v},\mathbf{a})\sim(T,V,A)}[y \log p_{dtm} + (1-y)\log(1-p_{dtm})], \tag{7}$$

in which $p_{dtm}$ are the output probabilities when feeding again caption tokens into the text encoder activating cross-attention layers to attend to the concatenated (along the sequential dimension) audio and video features as conditioning. Summing up everything, the final loss function is:

$$\mathcal{L}_{TOT} = \frac{1}{2}(\mathcal{L}_{D2T} + \mathcal{L}_{T2D}) + \lambda\mathcal{L}_{DTM}. \tag{8}$$

**Downstream Tasks Regularization.** To further enhance the performance of TRIANGLE in multimodal alignment and effectively manage all possible vector positions in the higher-dimensional space, we propose to add a regularization to the area minimization process in downstream tasks. This regularization leverages a two-dimensional cosine similarity to complement the proposed alignment strategy with an existing lower-dimensional similarity measure. In this way, area minimization ensures the contribution of all the modalities, while cosine regularization specifically controls only the alignment of the most relevant modalities to the downstream task. This dual approach guarantees true alignment in space, even in exceptional cases. A visual example is shown in Fig. 3 (c).

Formally, let $\mathbf{x}, \mathbf{y}, \mathbf{z}$ be the three modality embeddings, and define $\mathbf{u} = \mathbf{x} - \mathbf{y}$, $\mathbf{v} = \mathbf{x} - \mathbf{z}$ as the sides of the triangle. In a retrieval downstream task between the modalities $\mathbf{x}$ and $\mathbf{y}$, whose angle is $\theta_{\mathbf{xy}}$, the proposed alignment strategy can be expressed as:

$$\mathcal{A} = \frac{1}{2}\sqrt{\langle \mathbf{u}, \mathbf{u}\rangle\langle \mathbf{v}, \mathbf{v}\rangle - \langle \mathbf{u}, \mathbf{v}\rangle^2} - \alpha\cos\theta_{\mathbf{xy}}, \tag{9}$$

where the first term on the right side represents the area of the triangle formed by the embeddings and $\alpha$ is the regularizing hyperparameter that balances the contribution of the area minimization and the cosine similarity in the alignment strategy.

This formulation ensures that the area minimization captures the alignment across all three modalities, while the cosine similarity regularization fine-tunes the alignment specifically for the retrieval task between $\mathbf{x}$ and $\mathbf{y}$.

### 3.4 Advantages of the TRIANGLE

Firstly, TRIANGLE effectively models three modalities directly in the higher-dimensional space, without the need to employ some fusion strategies for any two of them. By avoiding the use of fusing layers for multiple modalities, TRIANGLE can work with different modalities: regardless of which modalities are involved, they will always define a triangle for which the area can be computed. This is a crucial advantage that makes the proposed method both portable and versatile across different types of data. Moreover, the TRIANGLE objective offers an intuitive indicator of how closely the three embeddings cluster: a low value of the area $A$ indicates that the embeddings are well-aligned, while a high value suggests they are far apart. Indeed, in the Appendix, we show how the area value among matching embeddings decreases during training. These geometric constraints are pivotal when performing downstream tasks on unseen datasets, thus ensuring the effective alignment of modalities. In practice, the true alignment of the three modalities directly improves the model performance, particularly with sample data in which the information contained in the third modality is essential to solve the task, as shown in Fig. 1. Finally, in terms of computational load, TRIANGLE brings negligible increment of the computational time with only 0.0016 seconds to compute the area of three vectors of dimension 2048 against the 0.0001 seconds of the cosine similarity computation with a batch of size 256 on an RTX4080 in inference.

Table 1: Zero-shot multimodal text-to-video (T2V) and video-to-text (V2T) retrieval R@1 results. Increment points computed wrt VAST with same modalities, number of parameters, and encoders. The difference between VAST and TRIANGLE results is statistically significant ($p < 0.001$).

| | Modality | MSR-VTT | | DiDeMo | | ActivityNet | | VATEX | |
|---|---|---|---|---|---|---|---|---|---|
| | | T2V | V2T | T2V | V2T | T2V | V2T | T2V | V2T |
| UMT Liu et al. (2022) | T-V | 33.3 | - | 34.0 | - | 31.9 | - | - | - |
| OmniVL Wang et al. (2022a) | T-V | 34.6 | - | 33.3 | - | - | - | - | - |
| UMT-L Li et al. (2023) | T-V | 40.7 | 37.1 | 48.6 | 49.9 | 41.9 | 39.4 | - | - |
| TVTSv2 Zeng et al. (2023) | T-V | 38.2 | - | 34.6 - | - | - | - | - | - |
| ViCLIP Wang et al. (2023) | T-V | 42.4 | 41.3 | 18.4 | 27.9 | 15.1 | 24.0 | - | - |
| VideoCoCa Yan et al. (2022) | T-V | 34.3 | 64.7 | - | - | 34.5 | 33.0 | 53.2 | 73.6 |
| Norton Lin et al. (2024) | T-V | 10.7 | | - | - | - | - | - | - |
| ImageBind Girdhar et al. (2023) | T-V | 36.8 | - | - | - | - | - | - | - |
| InternVideo-L Wang et al. (2022b) | T-V | 40.7 | 39.6 | 31.5 | 33.5 | 30.7 | 31.4 | 49.5 | 69.5 |
| HiTeA Ye et al. (2022) | T-V | 34.4 | - | 43.2 | - | - | - | - | - |
| mPLUG-2 Xu et al. (2023) | T-V | 47.1 | - | 45.7 | - | - | - | - | - |
| VideoPrism-b Zhao et al. (2024) | T-V | 51.4 | 50.2 | - | - | 49.6 | 47.9 | 62.5 | 77.1 |
| LanguageBind Zhu et al. (2024) | T-V | 44.8 | 40.9 | 39.9 | 39.8 | 41.0 | 39.1 | - | - |
| GRAM Cicchetti et al. (2025) | T-VA | 54.2 | 50.5 | 54.2 | 52.2 | 59.0 | 50.4 | **83.9** | 79.2 |
| VAST Chen et al. (2023b) | T-VA | 49.3 | 43.7 | 49.5 | 48.2 | 51.4 | 46.8 | 80.0 | 77.3 |
| TRIANGLE (ours) | T-VA | **55.2** | **52.5** | **54.9** | **53.1** | **59.7** | **54.1** | **83.9** | **80.9** |
| *TRIANGLE Improvement wrt VAST* | | +5.9 | +8.8 | +5.4 | +4.9 | +8.3 | +7.3 | +3.9 | +3.6 |

# 4   Experiments

We run experiments on seven popular benchmarks and we conduct three different types of evaluation: first, a vanilla experiment in a controllable environment, then extensive experiments with pertaining and downstream tasks, and finally the training from scratch.

## 4.1   Vanilla Experiment

Let us consider an experiment in a controllable environment with three modalities and a latent space of dimension 3. We build a framework comprising three modalities: images with the MNIST dataset, audio with the AudioMNIST dataset Becker et al. (2023), and text, with the textual labels associated with the digit numbers from 0 to 9. The objective is correctly retrieving the label from the image and audio representations. We develop three vanilla encoders, a two-layer convolutional encoder for images, a three-layer convolutional encoder for audio spectrograms, and Word2Vec for text data. We run trainings with different losses: the conventional pairwise cosine-based loss, therefore computing the cosine similarity between text and video, and then text and audio, selecting as anchor the text like in Zhu et al. (2024); the Symile Saporta

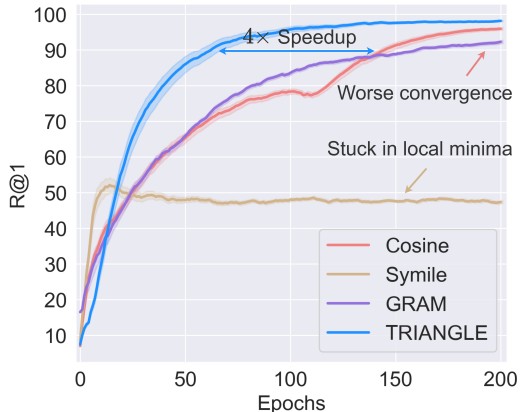

Figure 4: The proposed TRIANGLE shows a faster and better convergence overall.

et al. (2024) loss function, the GRAM one Cicchetti et al. (2025), and the proposed TRIANGLE in (5) and (6). Figure 4 shows the results of the experiment. The proposed TRIANGLE obtains a better, smoother, faster convergence, up to $4\times$ speedup to reach 90% than the conventional cosine-based loss and than GRAM, while achieving the best R@1 score overall. Symile, instead, is stuck in a local minimum that prevents it from surpassing an R@1 higher than 50. These results show the superior performance of TRIANGLE in better shaping a unified latent space to favor the alignment of all the modalities, leading to improved retrieval performance.

### 4.2 Extensive Experimental Evaluation

#### 4.2.1 Settings

The TRIANGLE model exploits the novel contrastive loss functions defined in (5) and (6) to fully align three modalities in a joint fashion. For the encoder models, we select as backbone the SOTA model designed for multiple modalities, VAST Chen et al. (2023b), thus employing BERT-B, BEATs Chen et al. (2023c), and EVAClip-ViT-G Sun et al. (2023) as text, audio, and video encoder, respectively. We pretrain the TRIANGLE model on top of VAST Chen et al. (2023b) (removing the fusing layers) on a subset of 150k samples randomly selected from the VAST27M dataset Chen et al. (2023b). We also applied the additional pretraining strategy on 150k samples to VAST, but this model quickly overfits, bringing no improvements. This is probably due to the convergence that the VAST model reached on the whole VAST27M dataset. On the contrary, TRIANGLE is able to reshape and remodel the latent space learned by VAST encoders towards a more aligned and unified multimodal space. Following the experimental receipts of existing embedding models in the literature, we validate the TRIANGLE performance on three fundamental downstream zero-shot tasks: video retrieval (text-to-video and video-to-text), audio retrieval (text-to-audio), and audio-text classification. Superior performance on these tasks suggests a better alignment of the various modalities within the shared embedding space, highlighting the effectiveness of the proposed TRIANGLE method.

#### 4.2.2 Zero-Shot Video Retrieval

The video retrieval task can be divided into two subtasks: 1) *Text-to-Video Retrieval (T2V):* Given a natural language query, find the most relevant video from a set of candidate videos. 2) *Video-to-Text Retrieval (V2T):* Given a candidate video, find the most relevant natural language query.

We evaluate TRIANGLE zero-shot on a wide range of popular benchmarks, namely MSR-VTT Xu et al. (2016), DiDeMo Hendricks et al. (2017), ActivityNet Caba Heilbron et al. (2015) and VATEX Wang et al. (2019), as shown in Tab. 1 for video-text tasks. These datasets contain the three modalities of interest (i.e., video frames, audio waves, and video captions). For each dataset, we use 8 video frames randomly selected, as in Chen et al. (2023b). More details about dataset sizes and hyperparameters in the Appendix. We report R@1 scores for video-to-text and text-to-video results in Tab. 1. The results in Tab. 1 uniquely demonstrate the superiority of TRIANGLE against established SOTA models, both vision-only and multimodal models. Indeed, both in V2T and T2V tasks, TRIANGLE obtains the best performance overall. Spefically, with respect to VAST Chen et al. (2023b), which has the same encoders, number of parameters, and pretraining dataset, TRIANGLE improves the performance up to 9 points and overall by a minimum of 4 points of R@1. Such a breakthrough result is due to the ability of TRIANGLE to effectively leverage the third modality, thus leading to enhanced performance. Indeed, the audio modality introduces more information essential for solving the text-to-video task, especially in video with a strong audio component that is reflected in the caption but that cannot be exploited solely from the frames. While VAST also uses three modalities, it fuses visual and audio embeddings using an MLP before comparing them to text via cosine similarity. This approach does not provide any geometric indicator of the alignment, thus limiting its effectiveness in fully aligning the modalities. Instead, TRIANGLE improves this approach by using the straightforward formulation in (9) incorporated in the brand-new contrastive losses in (5), which shows a clear geometric explanation and ensures a more robust alignment of the modalities. Furthermore, TRIANGLE outperforms generic $n$-modal methods like GRAM Cicchetti et al. (2025) and Symile Saporta et al. (2024), proving that the proposed similarity tailored for three modalities can better exploit three-modal cues than fully general $n$-modal losses.

#### 4.2.3 Zero-Shot Audio Retrieval and Classification

We evaluate TRIANGLE in zero-shot audio retrieval on two popular benchmarks: AudioCaps Kim et al. (2019) and VGGSound Chen et al. (2020). On Audiocaps we compute the zero-shot text-to-audio retrieval task. On VGGSound we compute zero-shot audio-visual classification. Due to the new YouTube policies, the complete VGGSound test set is not available anymore, thus we compute all the metrics (including comparisons) on a subset comprising 5k samples. Results for both the datasets and tasks are reported in Tab. 2 in terms of Recall@1 and Recall@10. TRIANGLE clearly outperforms all previous methods by up to 12 points, thus establishing new state-of-the-art results across both examined datasets and tasks. Such a large improvement is due to TRIANGLE effective leverage of the third modality. Including the video modality introduces crucial information essential

Table 2: Zero-shot text-to-audio **retrieval** results on AudioCaps and audio-text **classification** results on VGGSound 5K.

| | Modality | AudioCaps | | VGGSound 5K | |
|---|---|---|---|---|---|
| | | R@1 | R@10 | R@1 | R@10 |
| AVFIC Nagrani et al. (2022) | T-A | 8.7 | 37.7 | - | - |
| AVFIC Nagrani et al. (2022) | T-AV | 10.6 | 45.2 | - | - |
| VIP-ANT Zhao et al. (2022) | T-A | 27.7 | 37.7 | - | - |
| ImageBind Girdhar et al. (2023) | T-A | 9.3 | 42.3 | - | - |
| LanguageBind Zhu et al. (2024) | T-A | 19.7 | 67.6 | 23.8 | 57.1 |
| LanguageBind Zhu et al. (2024) | T-V | - | - | 37.2 | 62.0 |
| VAST Chen et al. (2023b) | T-V | - | - | 38.7 | 72.8 |
| VAST Chen et al. (2023b) | T-A | - | - | 25.6 | 56.2 |
| GRAM Cicchetti et al. (2025) | T-AV | **33.2** | 75.3 | 40.6 | 78.1 |
| VAST Chen et al. (2023b) | T-AV | 32.1 | 65.4 | 39.6 | 74.5 |
| TRIANGLE (ours) | T-AV | **32.2** | **77.1** | **44.8** | **80.0** |
| *Improvement wrt VAST* | | +0.1 | +11.7 | +5.2 | +5.5 |

for solving the audio retrieval task, particularly for captions that are challenging to infer using the audio modality alone. TRIANGLE obtains improved scores also in audio-visual classification, still boosting the performance by more than 5 points in both R@1 and R@10.

### 4.3 Learning the Space from Scratch

We perform a deeper study on the ability of TRIANGLE to better model the latent space by letting TRIANGLE losses learn from scratch on the MSR-VTT dataset for the multimodal text-to-audio/video (T2AV) and audio/video-to-text (AV2T) tasks. In this scenario, we select the same three encoders and then compare VAST (pairwise cosine similarity-based with fusing MLP layers) Chen et al. (2023b), Symile Saporta et al. (2024) as it leverages the total correla-

Table 3: Training from scratch and ablation study on MSR-VTT, with same encoders and different loss functions (cosine, Symile, GRAM), and the proposed TRIANGLE method w/ and w/o the $\mathcal{L}_{DTM}$. Statistically significant at $p < 0.05$.

| Training from scratch | T2AV | | AV2T | |
|---|---|---|---|---|
| | R@1 | R@10 | R@1 | R@10 |
| VAST Chen et al. (2023b) | 36.5 | 79.3 | 35.5 | 77.3 |
| Symile Saporta et al. (2024) | 0.3 | 3.1 | 0.4 | 3.6 |
| GRAM Cicchetti et al. (2025) | 38.9 | 80.8 | **41.9** | 79.5 |
| TRIANGLE w/o $\mathcal{L}_{DTM}$ | 33.3 | 74.4 | 40.4 | **81.7** |
| TRIANGLE (ours) | **39.4** | **81.8** | **41.9** | 80.0 |

tion to align multiple modalities, GRAM Cicchetti et al. (2025) that can align $n$ modalities through volume, and the proposed TRIANGLE loss function. These models are the sole ones to effectively perform multimodal tasks, although TRIANGLE is the only one tailored for the specific three-modal case. The encoders possess no pretrained knowledge in this experiment, and the four methods are left free to shape the latent space according to the loss minimization. Table 3 reports R@1 and R@10 scores that explicitly highlight the superior performance of TRIANGLE. Despite the greater generality, Symile and GRAM underperform TRIANGLE on all three-modal retrieval benchmarks. Once again, this suggests that an objective tailored to triplets can exploit modality-specific cues more effectively than fully general $n$-modal losses.

**Ablation Study.** We perform an ablation study on the effect of the $\mathcal{L}_{DTM}$ loss function in TRIANGLE. From Tab. 3, it is clear that the usage of such a loss function is crucial to obtaining better performance, and the proposed TRIANGLE configuration achieves the best scores overall. We perform more ablation studies in the Appendix.

## 5 Conclusion

We introduced TRIANGLE: TRI-modAl Neural Geometric LEarning, a novel method for the alignment of three modalities. Addressing the challenges of modality alignment that limit the performance of SOTA multimodal models, TRIANGLE introduces a novel similarity measure computed directly in the higher-dimensional space spanned by the modalities embeddings. This promotes joint alignment across the three studied modalities without additional anchors or additional fusion layers. Our extensive evaluations demonstrate that TRIANGLE not only enhances the interpretability of the alignment process but also reaches zero-shot SOTA performance in multimodal downstream tasks.

## Acknowledgments

This work was partially supported by the European Union under the NRRP of NextGeneration-EU, partnership on "Future Artificial Intelligence Research" (PE00000013 – SPOKE 5 - CUP B53C22003980006 - FAIR: High Quality AI) and partnership on "Telecommunications of the Future" (PE00000001 - program "RESTART"), partially by the *Progetti di Ateneo* of Sapienza University of Rome under grant RM123188F75F8072 and RM1241910FC4BEEA, and partially by the Italian Ministry of University and Research (MUR) within the PRIN 2022 Program for the project "EXEGETE: Explainable Generative Deep Learning Methods for Medical Signal and Image Processing", under grant number 2022ENK9LS, CUP B53D23013030006.

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

# Appendix

## 5.1 Area of the TRIANGLE in any dimension

**Definition 5.1** (**The TRIANGLE area is a measure of similarity**). The area $A$ of a triangle in $\mathbb{R}^n$ is given by:

$$A = \frac{1}{2}\sqrt{\langle \mathbf{u}, \mathbf{u} \rangle \langle \mathbf{v}, \mathbf{v} \rangle - \langle \mathbf{u}, \mathbf{v} \rangle^2}, \tag{10}$$

*Proof.* Prerequisites:

1. $\langle \mathbf{x}, \mathbf{y} \rangle = \mathbf{x} \cdot \mathbf{y} = x_1 y_1 + \cdots + x_n y_n; \mathbf{x}, \mathbf{y} \in \mathbb{R}^n$

2. $||\mathbf{x}|| = \sqrt{\langle \mathbf{x}, \mathbf{x} \rangle}$

3. $\sin^2 \theta + \cos^2 \theta = 1; \qquad \sin \theta = \pm\sqrt{1 - \cos^2 \theta}$

4. $\cos \theta_{xy} = \frac{\langle \mathbf{x}, \mathbf{y} \rangle}{||\mathbf{x}|| ||\mathbf{y}||} = \frac{\langle \mathbf{x}, \mathbf{y} \rangle}{\sqrt{\langle \mathbf{x}, \mathbf{x} \rangle}\sqrt{\langle \mathbf{y}, \mathbf{y} \rangle}}$

Let us consider a generic triangle $\widehat{\mathbf{xyz}}$ with vertex $\mathbf{x}, \mathbf{y}, \mathbf{z} \in \mathbb{R}^n$. Let us define $\mathbf{u} = \mathbf{x} - \mathbf{y}$ and $\mathbf{v} = \mathbf{x} - \mathbf{z}$ as two adiacent triangle side. Let $\theta_{uv}$ be the angle formed by $\mathbf{u}$ and $\mathbf{v}$. Considering the prerequisites, the area of the triangle is defined as follows:

$$\begin{aligned}
A &= \frac{1}{2}\overline{\mathbf{xy}} \cdot \overline{\mathbf{xz}} \cdot \sin \theta_{uv} \\
&= \frac{1}{2}\sqrt{\langle \mathbf{u}, \mathbf{u} \rangle} \cdot \sqrt{\langle \mathbf{v}, \mathbf{v} \rangle} \cdot \sin \theta_{uv} \\
&= \frac{1}{2}\sqrt{\langle \mathbf{u}, \mathbf{u} \rangle \cdot \langle \mathbf{v}, \mathbf{v} \rangle} \cdot \sqrt{1 - \cos^2 \theta_{uv}} \\
&= \frac{1}{2}\sqrt{\langle \mathbf{u}, \mathbf{u} \rangle \cdot \langle \mathbf{v}, \mathbf{v} \rangle - (\langle \mathbf{u}, \mathbf{u} \rangle \cdot \langle \mathbf{v}, \mathbf{v} \rangle) \cdot \cos^2 \theta_{uv}} \\
&= \frac{1}{2}\sqrt{\langle \mathbf{u}, \mathbf{u} \rangle \cdot \langle \mathbf{v}, \mathbf{v} \rangle - (\langle \mathbf{u}, \mathbf{u} \rangle \cdot \langle \mathbf{v}, \mathbf{v} \rangle) \cdot \frac{\langle \mathbf{u}, \mathbf{v} \rangle^2}{\langle \mathbf{u}, \mathbf{u} \rangle \cdot \langle \mathbf{v}, \mathbf{v} \rangle}} \\
&= \frac{1}{2}\sqrt{\langle \mathbf{u}, \mathbf{u} \rangle \cdot \langle \mathbf{v}, \mathbf{v} \rangle - \langle \mathbf{u}, \mathbf{v} \rangle^2}.
\end{aligned}$$

$\square$

## 5.2 Experiments Details

We perform both pretraining and training from scratch of the TRIANGLE model using $4 \times$ A100 GPUs. For pretraining, we sample 150,000 examples from the VAST27M dataset. Each example consists of a video (comprising frames and an audio track) paired with a corresponding caption. The model backbone is adapted from VAST Chen et al. (2023b), utilizing BERT-B, BEATs Chen et al. (2023c), and EVA-CLIP ViT-G Sun et al. (2023) as the text, audio, and video encoders, respectively, resulting in a total of 1.3 billion parameters.

Following the approach in Chen et al. (2023b), for the generic $i - th$ sample, we extract sparsely sampled video frames from it and pass it through the video encoder to obtain the video representation $t_i$. The caption text is tokenized and processed by the text encoder to produce text embeddings $t_i$. For the audio modality, the audio track is segmented into 10-second clips, zero-padded as necessary, converted into a 64-dimensional log Mel filterbank spectrogram using a 25ms Hamming window, and encoded via the audio encoder to yield the audio representation $a_i$.

During pretraining phase, we use just one single frame and a single 10-second audio clip extracted from each sample; In the training-from-scratch setting, we use eight randomly selected video frames and a single 10-second audio clip per video.

Table 4: Dataset statistics and hyperparameters. Modalities stand for T: text, V: video, A: audio. # F refers to the number of frames used for testing phase. # AC refers to the number of 10-seconds long audio clips used during testing phase

| Benchmark | #Video / #Audio | | # F | # AC |
|-----------|-------|------|-----|------|
| | Train | Test | | |
| MSR-VTT | 9000 | 1000 | 8 | 1 |
| DiDeMo | - | 1003 | 8 | 1 |
| ActivityNet | - | 4917 | 8 | 1 |
| VATEX | - | 431 | 8 | 1 |
| AudioCaps | - | 700 | 8 | 1 |
| VGGSound | - | 5000 | 8 | 1 |

TRIANGLE is pretrained for 10k steps on the 150k-sample subset of VAST27M. Retrieval performance is evaluated every 100 steps on the MSR-VTT test set, and the checkpoint with the best performance is selected. We employ an initial learning rate of 1e-4 with a linear decay schedule and a batch size of 256. In the training-from-scratch experiments we train from scratch the aforementioned encoders on the MSRVTT train dataset for 4 epochs with an initial learning rate of 1e-4 with a linear decay schedule and a batch size of 64.

To evaluate the performance of TRIANGLE, we follow a standard evaluation protocol, randomly selecting eight video frames and one 10-second audio clip from each video to enable fair comparison with existing state-of-the-art models.

We utilize several benchmark datasets for our downstream tasks:

**MSR-VTT** Xu et al. (2016) contains 10,000 video clips paired with 200,000 captions, covering a broad range of topics such as human activities, sports, and natural landscapes.

**DiDeMo** Hendricks et al. (2017) includes 10,000 long-form videos from Flickr, each annotated with four temporally ordered natural language descriptions. These captions correspond to distinct moments within each video, offering fine-grained alignment between textual and visual content.

**ActivityNet** Caba Heilbron et al. (2015) comprises 20,000 long-form YouTube videos (averaging 180 seconds in duration) and 100,000 captions. It spans 200 human activity classes, from daily routines to complex sports and interactions. Each video is annotated with both activity labels and temporal boundaries, enabling precise localization of actions.

**VATEX** Wang et al. (2019) consists of 41,250 video clips sourced from the Kinetics-600 dataset **?**, accompanied by 825,000 sentence-level descriptions. However, due to a significant portion of videos becoming unavailable online (e.g., removed or made private), we use a curated subset of 14,491 samples.

**AudioCaps** Kim et al. (2019) consists of 51,000 audio clips, each 10 seconds long. The training set includes one caption per clip, while the validation and test sets provide five captions per clip. We follow the dataset split protocol proposed by **?** for the text-to-audio retrieval task.

**VGGSound** Chen et al. (2020) is a large-scale audio-visual dataset with over 200,000 YouTube video clips, each 10 seconds long and labeled with one of 309 audio classes. The dataset covers a wide array of sound events, including human actions, animal sounds, environmental noises, and mechanical events. Due to download limitations, we use a subset of 5,000 samples for testing.

### 5.3 Visualizing Learning Curves

Figure 5 shows the loss functions for the vanilla experiment, in which it is clear that the proposed TRIANGLE obtains a better convergence with respect to the conventional cosine-based loss and to Symile and GRAM.

Furthermore, we also plot the area value among true matching pairs against the R@1 in the training from scratch experiment on MSR-VTT. Figure 6 shows the results. As it is evident, the area value is minimized during training and, concurrently, the R@1 increases as the area among the true pairs decreases.

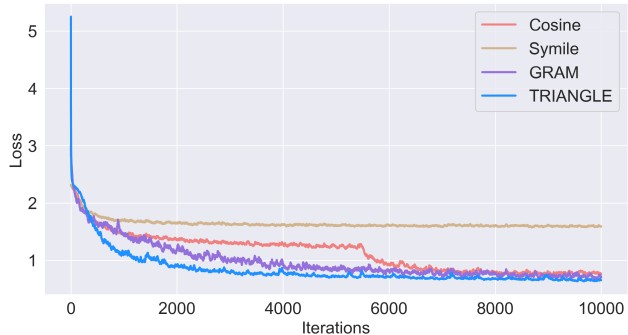

Figure 5: Vanilla experiment losses.

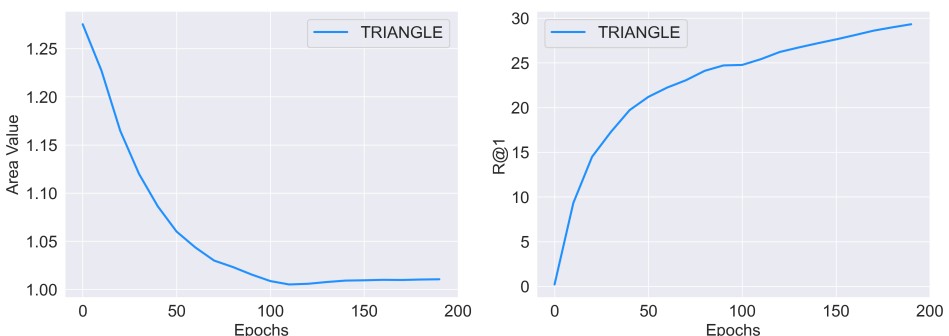

Figure 6: Area value versus R@1 during training from scratch on MSR-VTT dataset.

## 5.4 Examples of Retrieval Outputs

We briefly describe the outcomes from three representative multimodal examples.

**Sample 1** `VGGSOUND ID: -U7joUcTCo` from $T = 0s$ (i.e., from second 0 of the video)
Ground truth class: *People coughing*

**Results for VAST**
Rank of ground truth class: **22nd**
TOP 5 Retrieved classes: [people belly laughing, people giggling, people whistling, male speech man speaking, people cheering]

**Results for TRIANGLE**
Rank of ground truth class: **3rd**
TOP 5 Retrieved classes: [people giggling, people belly laughing, **people coughing**, baby laughter, people sobbing]

**Sample 2** `VGGSOUND ID: -0vPFx-wRRI` from $T = 30s$
Ground truth class: *People finger snapping*

**Results for VAST**
Rank of ground truth class: **22nd**
TOP 5 Retrieved classes: [male singing, child speech kid speaking, child singing, playing castanets, people giggling]

**Results for TRIANGLE**
Rank of ground truth class: **1st**
TOP 5 Retrieved classes: [**people finger snapping**, people eating apple, male singing, playing castanets, people eating crisps]

**Sample 3**  `VGGSOUND ID: 2YoOMGR2kmA`   from $T = 31s$
Ground truth class: *Playing bagpipes*

**Results for VAST**
Rank of ground truth class: **30th**
TOP 5 Retrieved classes: [playing ukulele, playing sitar, playing mandolin, playing steel guitar slide guitar]

**Results for TRIANGLE**
Rank of ground truth class: **1st**
TOP 5 Retrieved classes: [**playing bagpipes**, playing ukulele, playing banjo, playing sitar, playing steelpan]

From our qualitative analysis, we consistently observe that TRIANGLE effectively integrates information from all three modalities (visual, audio, and text). This results in superior performance on tasks such as video classification and reasoning, where a holistic understanding across modalities is crucial. A particularly illustrative example is Sample 3 from our previous analysis. In the video, a man holding a guitar or ukulele appears in the foreground, while another man with a bagpipe is seen in the background. However, upon listening to the audio, it becomes clear that only the bagpipe is being played. TRIANGLE successfully captures this multimodal context and correctly classifies the video as "Playing bagpipes", whereas VAST misclassifies it as "Playing ukulele" by likely over-relying on visual cues.

## 5.5  Experiments with Different Modalities

We test TRIANGLE on a different set of modalities in the Touch-Vision-Language Dataset Fu et al. (2024), which comprises in-the-wild vision-touch pairs with language labels. The current state-of-the-art model, Touch-Vision-Language (TVL), leverages pairwise cosine similarity among all the pairwise, training the tactile encoder to align it to the vision and text encoder from CLIP. We retrained TVL (first row in the table) and we train the same models with the proposed TRIANGLE loss (second row in the table). As it is clear from the scores in Tab. 5, TRIANGLE outperforms TVL in the standard two-modal task, better aligning vision and tactile modalities. What is more, by involving our three-modal loss, we can also unlock a novel task, Vision to Tactile&Text, that allows us to effectively align the three modalities altogether, building a joint latent space.

Table 5: Comparison of TVL model Fu et al. (2024) on the TVL dataset against the proposed TRIANGLE, which outperforms the original TVL method.

|  | Acc@1 | Acc@5 | Vis.-Tact. Acc@1 | Vis.-Tact. Acc@5 | Vis.-Tact.-Text Acc@1 | Vis.-Tact.-Text Acc@5 |
|---|---|---|---|---|---|---|
| TVL | 36.7 | 53.3 | 79.9 | 93.1 | – | – |
| TRIANGLE | **83.1** | **94.6** | **82.8** | **94.6** | **83.5** | **94.7** |

## 5.6  Ablation Studies

### 5.6.1  Ablation Study on $\alpha$

We conduct ablation studies to validate the effectiveness of the TRIANGLE choices. In Fig. 7 we report ablation studies made on the regularization weight $\alpha$ of (9). We propose to add a regularization term to the area minimization using cosine similarity between the two modalities most relevant to the downstream task (i.e., for video-text retrieval tasks, the audio may be crucial for the downstream task, but the contribution of video and text are surely more important). The weight $\alpha$ is used to balance the contribution of area similarity and cosine similarity. If $\alpha = 0$, then only area is considered; if $\alpha = 1$, equal weight is given to area and cosine similarity. As shown in Fig. 7, due to the particular case shown in Fig. 3 (c) that cannot be properly interpreted by the area, it alone may produce suboptimal results. Injecting the cosine similarity regularization performance starts to improve. Interestingly, we discover that this improvement is far more meaningful in the Text to Visual-Audio retrieval task, whereas it is not so relevant in the visual/audio to text retrieval task. According to these ablations, we set $\alpha = 1$ for the first deck of experiments and $\alpha = 0$ for the latter one.

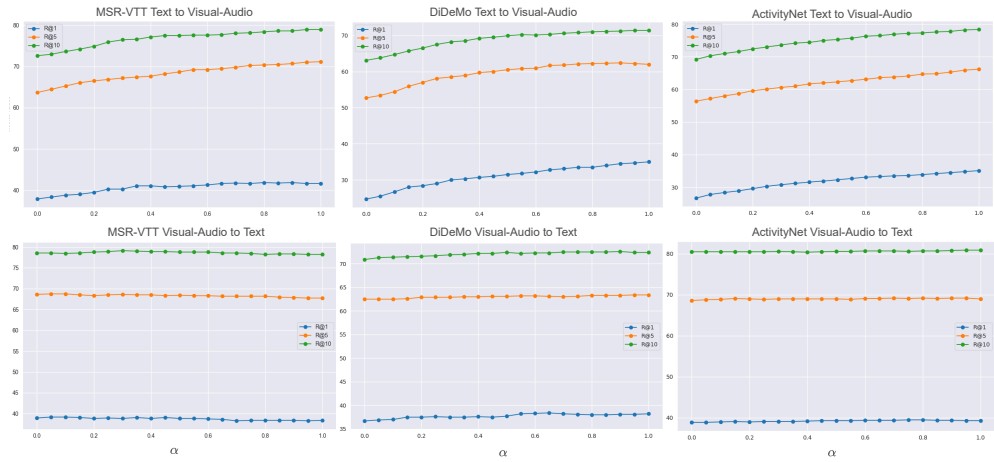

Figure 7: Ablation studies for weight regularization $\alpha$ on sample datasets MSR-VTT, DiDeMo, and ActivityNet.

Table 6: Performance of TRIANGLE trained from scratch on MSR-VTT under different values of $\lambda$.

|  | $\lambda$ | T2AV R@1 | T2AV R@10 | AV2T R@1 | AV2T R@10 |
|---|---|---|---|---|---|
| TRIANGLE | 0.0 | 33.3 | 74.4 | 40.4 | **81.7** |
| TRIANGLE | 0.1 | **39.4** | **81.8** | **41.9** | 80.0 |
| TRIANGLE | 0.3 | 20.9 | 71.6 | 20.7 | 72.4 |
| TRIANGLE | 0.5 | 24.0 | 74.2 | 22.2 | 71.9 |
| TRIANGLE | 1.0 | 38.3 | 81.2 | 36.9 | 78.4 |

### 5.6.2 Ablation Study on $\lambda$

We perform an additional ablation study on the value of $\lambda$ that regulates the influence of $L_{DTM}$ in the total loss in (8). In all the experiments conducted in the paper, we set $\lambda = 0.1$, inhereting this value from previous works (VAST and GRAM). In this ablation, we test different values ranging in (0.0, 0.1, 0.3, 0.5, 1.0). From the result table below, it is clear that the setting with $\lambda = 0.1$ obtains the best results in 3 over 4 scores, thus confirming the setting choice of the experiments in the main paper.

### 5.7 Limitations and Future Work

TRIANGLE is designed to solve three-modal tasks. However, its formulation can be also extended to more modalities. In the case of more than three modalities, such embeddings form a generic polygon. This polygon can be divided into smaller triangles for which we can straightforwardly compute the area with eq. (10) in the main paper. The resulting area will be the sum of the subtriangles area. Several ways to define the polygon can be potentially explored. Among others, we may leverage the Convex Hull that computes the smallest convex set containing all the points under consideration. We plan to investigate these solutions in future work.

