# OpenReview forum: "A TRIANGLE Enables Multimodal Alignment Beyond Cosine Similarity"
_NeurIPS.cc/2025/Conference — NeurIPS 2025 poster_

### Official Review · Reviewer_bcfQ · 2025-07-01

**Clarity:** 3
**Significance:** 2
**Originality:** 2
**Rating:** 5
**Confidence:** 3

**Summary:**

This work suggests a new loss function to align modalities in self-supervised multimodal models that take three modalities as input. The work begins by noting that standard cosine similarity, which only enables pairwise comparisons fails to directly optimize for the $n$-modality setting where $n > 2$. To address this, the paper proposes a new version of contrastive loss for the $n =3$ setting, TRIANGLE, that replaces calculation of cosine similarity with the area of the triangle generated by the three representations (one from each modality). The paper compares this function to existing approaches (both those that use an anchor modality and also broader generalizations that use notions of volume), noting that it adds limited computational overhead. Finally, the paper runs a range of experiments both in toy settings (multimodal MNIST) and in various more realistic retrieval tasks. Overall, TRIANGLE outperforms comparable approaches in many cases.

**Questions:**

- Line 338: The paper finds that the DTM score is essential. Can more be said about why this is?
- Have the authors considered trying to extend TRIANGLE to more than 3 modalities?
- Why isn’t (5) symmetric in its indices?

**Ethical Concerns:**

["NO or VERY MINOR ethics concerns only"]

**Final Justification:**

The authors addressed most of my concerns during the rebuttal phase including
- Clearing up that the statement about anchors and alignment was primarily empirical in nature
- More clearly contrasting TRIANGLE to other similar approaches
- And adding another experiment.
Overall, I think that this is an interesting paper that is much improved by the additions made during revision.

**Limitations:**

The paper does not address limitations. Limitations to address might include the fact that this only works for exactly three modalities. Another limitation is that this method is mostly evaluated in terms of retrieval.

**Paper Formatting Concerns:**

The paper formatting seems fine.

**Quality:**

3

**Strengths And Weaknesses:**

Strengths:
- **The method is simple (in a good way) and makes intuitive sense:** TRIANGLE is intuitive, leveraging basic geometric concepts like area that hold even in high-dimensional spaces. Adding to its appeal, it would also be easy to integrate it into existing multimodal training or finetuning frameworks.
- **The paper is clearly written and easy to read:** Even though this reviewer does not work in multimodal contrastive learning, they found this paper easy to read. The prose is clear and the main points and analysis are made concisely.
- **At least for the tasks on which it is evaluated, TRIANGLE demonstrates strong performance:** In most cases, TRIANGLE achieves high performance relative to competing methods (but see some questions about this in the ‘Questions’ section below).


Weaknesses
- **The value of TRIANGLE is a bit more nuanced than what is conveyed in the paper and its relationships to similar methods are underemphasized:** One of the main points of the paper is that TRIANGLE overcomes a basic obstacle that many other approaches to multimodal contrastive learning have to grapple with: how to align more than two modalities via only pairwise comparisons. The paper does this by generalizing cosine similarity to the area of the triangle formed by the representation of the three modalities. The benefit of this approach is further emphasized in Sections 3.3 and 3.4 relative to anchor modality-based methods.  On the other hand, the paper also notes that there are at least two existing works that already address this through various generalizations of cosine similarity (e.g., parallelepiped volume, etc.). The reviewer would have liked to see a methodological comparison against these as TRIANGLE is a variation on at least GRAM. For instance, to this reviewer’s understanding, there should be nice comparisons of TRIANGLE to GRAM in the $n = 3$ setting.
- **The value of the MNIST experiments is unclear:** Small-scale toy experiments can be useful for exploring aspects of a problem that are challenging to run or analyze at scale. However, unless the reviewer missed something, the kinds of questions asked in Section 4.1 were more or less the same ones asked for the larger-scale experiments. The reviewer suggests either utilizing this section to ask questions appropriate for small scale settings or removing it.
- **Some of the claims about alignment appear wrong:** The paper claims that “this [anchor-based] alignment only ensures that all modalities are aligned with the anchor, without providing any indicators for the alignment between non-anchor modalities”. To this reviewer’s understanding, this is not true. Indeed, it is quite easy to see that when the pairwise comparison method is based on a true metric so that the triangle inequality holds, then when $d(a,b)$ and $d(a,c)$ are small, so is $d(b,c)$. The authors should correct me if I have misunderstood the claim here.

Nitpicks
- There are multiple places in the paper where ‘receipt’ should be ‘recipe’ (e.g., Line 28).
- The abstract is stilted and could use some polishing.
- Line 23: The notation here is confusing. What is $\mathbf{M}_i$? Given that $e_i: M_i \rightarrow \mathbf{M}_i$ one would assume that $\mathbf{M}_i$ is a set but then it is stated that $\mathbf{M}_i \in \mathbb{R}^n$, which doesn’t make sense.

---

> ### Author Rebuttal · Authors · 2025-07-30
>
> **On the comparison with GRAM.** We would like to thank the Reviewer for raising this interesting question. Let us focus on the three-modal case $n=3$ with text, audio, and vision, although the same analysis can be applied to any other set of modalities.
>
> Each method involves a unimodal encoder for each modality, and the normalized extracted embeddings are $\textbf{t}$ text, $\textbf{v}$ vision, and $\textbf{a}$ audio. We report only L_D2T; the same holds for L_T2D. In what follows, $\langle \cdot, \cdot \rangle$ denotes the dot product between two vectors of the same dimensionality.
>
> **Symile** formulates the loss function to ensure that all its components contribute to maximizing a lower bound on the total correlation among all modalities involved in the training. It relies on the multilinear inner product (MIP), which is not a measure of the geometric representations similarity, as stated in their paper. We found that, in real-world scenarios, Symile performance is weak, likely due to the lack of geometric principles behind their method. Formally, Symile loss can be defined as:
>
> $$L_{D2T\-Symile} =-\frac{1}{B}\sum_{i=1}^{B}\log\frac{\exp(\langle \textbf{t}_i, \textbf{v}_i, \textbf{a}_i \rangle /\tau)}{\sum^K_j\exp(\langle \textbf{t}_j, \textbf{v}_i, \textbf{a}_i \rangle/\tau)}$$
>
> Where $\langle \textbf{t}_i, \textbf{v}_i, \textbf{a}_i \rangle$ is the MIP, which is an extension of the classic dot product and is defined as $\langle \textbf{t}_i, \textbf{v}_i, \textbf{a}_i \rangle = \sum^D_d{\textbf{t}_i}_d \cdot {\textbf{v}_i}_d \cdot {\textbf{a}_i}_d$, with $D$ is the dimensionality of embeddings.
>
> **GRAM** introduces a geometrical approach for the multimodal pretraining of encoders based on the volume of the k-dimensional parallelotope spanned by the embeddings of the different modalities as
>
> $$L_{D2T\-GRAM}=-\frac{1}{B}\sum_{i=1}^{B}\log\frac{\exp(-Vol(\textbf{t}_i, \textbf{v}_i, \textbf{a}_i)/\tau)}{\sum^K_j\exp(-Vol(\textbf{t}_j, \textbf{v}_i, \textbf{a}_i)/\tau)}$$
>
> Where $Vol(\textbf{t}_i,\textbf{v}_i,\textbf{a}_i)= \sqrt{\det(\mathbf{G})}$; $\mathbf{G}=\mathbf{Z}^{\top}\mathbf{Z}$; $\mathbf{Z}=(\mathbf{t}_i,\mathbf{v}_i,\mathbf{a}_i)$. We will discuss the details of this method few lines below.
>
> **TRIANGLE** also is rooted on geometrical concepts, however, its formulation is different from GRAM. The area of the triangle built by the embeddings in the latent space is taken into consideration as an alignment metric of the embeddings themself, resulting in:
>
> $$L_{D2T\-TRIANGLE}=-\frac{1}{B}\sum_{i=1}^{B}\log\frac{\exp(-A(\textbf{t}_i, \textbf{v}_i, \textbf{a}_i)/\tau)}{\sum^K_j\exp(-A(\textbf{t}_j, \textbf{v}_i, \textbf{a}_i)/\tau)}$$
>
> here $A(\textbf{t}_i,\textbf{v}_i,\textbf{a}_i) = \frac{1}{2}\sqrt{\langle \mathbf{u}, \mathbf{u} \rangle \langle \mathbf{v}, \mathbf{v} \rangle - \langle \mathbf{u}, \mathbf{v} \rangle^2}$, in which  $\mathbf{u} = \mathbf{t}_i - \mathbf{a_i}$, $\mathbf{v} = \mathbf{t}_i - \mathbf{v}_i$ are two triangle sides computed among the three embeddings $\mathbf{t}_i, \mathbf{v}_i,$ and $\mathbf{a}_i$ of the three modalities.
> Although TRIANGLE can be mistakenly seen as a special case of GRAM when $n=3$, it can be shown that the two objective functions are fundamentally different mathematically and operate on distinct aspects of the geometrical alignment.
>
> Going into details, by expanding the volume formula and computing all the multiplication needed to compute the determinant of the matrix $\mathbf{G}$, it can be seen that
>
> $$Vol(\textbf{t}_i,\textbf{v}_i,\textbf{a}_i)= \sqrt{1 - \textbf{v}_i\textbf{a}_i^2 - \textbf{t}_i\textbf{a}_i^2 - \textbf{t}_i\textbf{v}_i^2 + 2 \cdot \textbf{t}_i\textbf{a}_i \cdot \textbf{a}_i\textbf{v}_i \cdot \textbf{v}_i\textbf{t}_i}$$
>
> Whereas the area of the triangle is:
>
> $$A(\textbf{t}_i,\textbf{v}_i,\textbf{a}_i) = \frac{1}{2}\sqrt{\langle (\mathbf{t}_i - \mathbf{a_i}), (\mathbf{t}_i - \mathbf{a_i})\rangle \langle (\mathbf{t}_i - \mathbf{v}_i), (\mathbf{t}_i - \mathbf{v}_i) \rangle - \langle (\mathbf{t}_i - \mathbf{a_i}), (\mathbf{t}_i - \mathbf{v}_i) \rangle^2}$$
>
> which is significantly different.
>
> Geometrically, the GRAM equation ($Vol$) is the volume of the parallelotope spanned by the three vectors with one vertex at the origin, measuring how mutually linearly independent the three origin-anchored vectors are. On the contrary, the TRIANGLE area ($A$) formula measures how far apart the  difference vectors are, i.e., the edges of the triangle with vertices at $\mathbf{t}, \mathbf{a}, \mathbf{v}$, not comprising the origin. Therefore, the two losses optimize different geometrical entities and rely on diverse formulations.
> These highlighted differences, when incorporated into contrastive learning loss functions, influence different aspects of the vector alignment.
>
> Experimental results demonstrate a clear improvement when using the TRIANGLE loss function compared to the GRAM loss function in the case of $n=3$ (results are reported in Tables 1, 2, and 3 of the paper). We highlight that all reported results for GRAM were obtained using the three-modal configuration ($n=3$) with T,V,A modalities like TRIANGLE to ensure a fair comparison.
>
> We are pleased to include this methodological analysis in the supplementary material of our work.
>
> **On the vanilla experiment.** We would like to thank the Reviewer for the suggestion. We moved the vanilla experiment on MNIST to the appendix and we inserted the experiment with different modalities (see response to Reviewer mfjz) in the main body, to further prove the robustness of TRIANGLE in different settings. Thanks for the suggestion!
>
> **On the pairwise alignment claims.** Thanks for giving us the opportunity to clarify this point. While there exists cases in which aligning the embeddings to an anchor yield the alignment of also non-anchor embeddings, there also exist cases in which this may not be true.
> Let us suppose to have three vectors $a,b,c$, where $a$ is the anchor, each with coordinates $a=(-1, 1, 0)$, $b=(-1,0,0)$, $d=(0,1,0)$. Then, $\theta=30°$ is the angle between $(a,b)$, which is quite tight, as $\cos(\theta)=0.87$ and reasonable to assume in a real-world scenarios. For simplicity, suppose also that the angle between $(a,c)$ is equal to $\theta$ as well. Therefore, both $b$ and $c$ are quite aligned with the anchor $a$. However, the angle between $b$ and $c$ themselves is equal to $2\theta$, and $\cos(2\theta)=0.5$, which instead is quite lower than the measure of alignment between embeddings and the anchor modality.
>
> Ineed, our claim is not that pairwise-to-anchor alignment cannot imply anything about the remaining pairs at all, but that it does not provide a direct, explicit indicator of that alignment, as the alignment of non-anchor embeddings is not explicit or measured in such methods and there may exist cases in which non-anchor pairs are indeed less aligned.
>
> **Typos and abstract** Thanks for pointing this out! We have revised them!
>
> **Notation** Thanks for pointing this out and sorry for this misleading notation, we have revised it.
> The role of encoders as a function is to transform data from their input domanin $\textit{M}_i$ into the representation space $\mathcal{Z}$, shared among all the modalities. Here, $\mathcal{Z}$ is a subset of $\mathbb{R}^n$ and so every vector $\mathbf{z}$ of $\mathcal{Z}$ is also a vector of $\mathbb{R}^n$.
> In conclusion $e_i: \textit{M}_i \rightarrow \mathcal{Z}$, $\mathcal{Z} \subseteq \mathbb{R}^n$.
>
>
> **Q1 On the role of DTM.** Following previous works such as VAST, to further fuse the multimodal embeddings, we use a multimodal encoder to rerank the pairs, outputing a score between 0 and 1 that is the probability of the three input embeddings have matching semantics or not. As explained in Sec 3.3, following a similar approach to VAST, we reuse the text encoder activating the cross attention layers to accomodate video and audio features to be a conditioning signal. This module is further trained using the loss L_DTM.
> We have added this clarification in the revised paper.
>
> **Q2 On extension to more than 3 modalities.** In Sec 5.5 Limitations and Future Work, we discuss potential ideas for extending the triangle area concept to the concept of area of a generic polygon in the hyperdimensional space that allows the extension of TRIANGLE to more than 3 modalities. Furthermore, we conduct preliminary experiments on the AV-MNIST dataset adding a fourth modality, that is the Colored MNIST, so to have a total set of 4 modalities, but the method can be generalized to more modalities. For details, please see Response to Reviewer u6BX on the Convex hull.
>
> **Q3 On the Symmetry of L_D2T** Our contrastive loss function 1/2(L_D2T + L_T2D) is rooted on many previous works relying on the InfoNCE loss function introduced in [1]. The InfoNCE is not symmetric: it optimizes only one directional alignment at a time, ignoring the opposite modality direction [2]. As a consequence, contrastive losses explicitly imply a second term (i.e. the reverse direction) to enforce symmetry and ensure consistent bidirectional alignment [3, LanguageBind, ImageBind, GRAM, VAST].
> Much more intuitivelly, with Eq 5 L_D2T many different text captions are contrasted with a single video-audio pair. For the video-audio pair that match with the correct textual description the corresponding area is reduced, for all the other triplets the area is enhanced.
> On the other hand, with Eq 6 L_T2D many different video-audio pairs are contrasted with a single textual description.
>
> [1] A. van den Oord, et al. "Representation learning with contrastive predictive coding" arXiv:1807.03748, 2018.
>
> [2] Lin, Zudi, et al. "Relaxing contrastiveness in multimodal representation learning" IEEE/CVF WACV. 2023.
>
> [3] A. Radford,  et al. "Learning transferable visual models from natural language supervision" ICML 2021.

---

> > ### Comment · Reviewer_bcfQ · 2025-08-03
> > **Reply to authors**
> >
> > We thank the authors for their clarifications. We believe that the comparisons with GRAM made in the beginning of the rebuttal will be particularly helpful.
> >
> > Unfortunately, we still don’t understand the argument around anchors. The authors provide a small example where two vectors $b$ and $c$ (I am assuming $d$ and $c$ are supposed to be the same vector and the extra name is just a typo) are pairwise orthogonal but are each closer to a third vector $a$. The reviewer may have misunderstood the set-up but given the values of $a$, $b$, and $c$, we find the angle between $a$ and $b$ and $a$ and $c$ to both be $45^\circ$ (the authors claim it is $30^\circ$). The fact that $b$ and $c$ can be orthogonal but $a$ and $b$ and $a$ and $b$ are not is just a consequence of the metric structure on the $n$-sphere. As the distances between $a$ and $b$ and $a$ and $c$ shrink, then the distance between $b$ and $c$ will shrink.
> >
> > If the point is that in practice, we can’t get the distance between $a$ and $b$ and $a$ and $c$ sufficiently small to pull $b$ and $c$ together, that is a more empirical point and should be explicitly stated that way.

---

> ### Author Response · Authors · 2025-08-04
> **Clarification on the pairwise alignment**
>
> We thank the Reviewer for the quick response.
> As the Reviewer correctly pointed out, the angle between $(a, b)$ is $45°$, and so the angle between $(a,c)$, resulting in an angle between $(b, c)$ of $90°$, we are sorry for the mistake. That was just a toy example to show a case in which $b$ and $c$ are quite aligned to the anchor but are instead misaligned between each other.
>
> Indeed, when using standard pairwise contrastive learning functions, things do not always work as intended. For example, the modality gap is a well-known phenomenon that occurs in every latent space: embeddings from different modalities tend to lie in separate regions of the space [1]. While the Reviewer's claim about the distances among three vectors is theoretically correct, in practice, the situation may be much more complex.
> We found that, in practice, matching pairs do not perfectly overlap and the angle between the anchor and the matching embedding of another modality does not shrink as much as necessary to build a properly aligned space among non-anchor modalities too. This limitation has been highlighted in [2] and in GRAM too, which explicitly state that current SOTA  fails to account for cross-modal correlation among non-anchor modalities.
> In our proposed loss function, we not only focus on the alignment between the anchor and other modalities, but also explicitly enforce alignment between the non-anchor modalities themselves. Moreover, differently from previous works that update weights of the model asyncroneously with one update for each modality pairs (see, for example, VAST, ImageBind, LanguageBind, ...), we perform the alignment whithin a single learning step considering the three modalities altogether.
>
> To prove our claims, we conduct an experiment on the MSR-VTT dataset. It reveals a weak alignment (around 1 and 5 points of R@1) of pairwise-based methods between non-anchor modalities. Instead, when jointly optimizing all the three modalities (text, video, audio) altogether with TRIANGLE, we can enhance the alignment between non-anchor modalities too.
>
> |              | V-A | A-V | T-VA        | VA-T        |
> |--------------|-----|-----|-------------|-------------|
> | LanguageBind | 1.1 | 1.6 | -           | 42.0        |
> | VAST         | 5.4 | 4.1 | 49.3        | 43.7        |
> | TRIANGLE     | 13.1    | 12.3    | 55.2 (+5.9) | 52.5 (+8.8) |
>
> We have revised the paper to better clarify this point and highlight our findings when exploring pairwise- and cosine-based methods. Thank you for pointing this out!
>
> [1] V. W. Liang, et al. "Mind the gap: Understanding the modality gap in multi-modal contrastive representation learning." Advances in Neural Information Processing Systems 35 (2022): 17612-17625.
>
> [2] M. Jeong, et al. “Anchors Aweigh! Sail for Optimal Unified Multi-Modal Representations.” ArXiv:2410.02086 2024.

---

> > ### Author Response · Authors · 2025-08-06
> >
> > Thank you again for your time and for your insightful and thorough review of our work.
> >
> > As the discussion period deadline is approaching, we wanted to respectfully follow up on our detailed rebuttal, where we sought to address every concern you raised.
> >
> > To summarize, in our responses we have:
> >
> > - Provided a methodological and mathematical comparison to previous works GRAM and Symile.
> >
> > - Clarified our claim regarding anchor-based alignment, supplementing it with an empirical study on MSR-VTT showing that TRIANGLE better aligns non-anchor modalities compared to pairwise approaches.
> >
> > - Clarified the rationale behind the DTM module and we have better explained the role of (L_D2T + L_T2D) losses.
> >
> > - Discussed possible extensions beyond the three-modality case.
> >
> >
> > We believe our detailed responses and the new experimental results thoroughly address the concerns you raised.
> > We would be very grateful for your feedback and are happy to answer any further questions.
> >
> > Thank you for your time and consideration.

---

> ### Comment · Reviewer_bcfQ · 2025-08-07
> **Reply to authors**
>
> We thank the authors for their clarification on the points raised.
>
> If the issue with anchors is primarily empirical, then this resolves this concern. The reviewer would ask that the paper's wording be modified to address this. To this reviewer at least, the original wording sounded like a fundamental mathematical claim which it seems that it is not.
>
> Given the changes, the reviewer is happy to raise their score to a 5.

---

### Official Review · Reviewer_tnky · 2025-07-01

**Clarity:** 3
**Significance:** 3
**Originality:** 3
**Rating:** 5
**Confidence:** 4

**Summary:**

This paper introduces TRIANGLE, a novel multimodal alignment approach that simultaneously aligns three modalities by minimizing the triangle area formed by their latent representations with a cosine regularization. The extensive experiments across multiple benchmarks demonstrate TRIANGLE's superiority compared to existing baselines on the downstream tasks (retrieval and classification).

**Questions:**

Please ref to **Weaknesses**

**Ethical Concerns:**

["NO or VERY MINOR ethics concerns only"]

**Final Justification:**

The authors largely addressed my concerns. I decide to update my score to 5.

**Limitations:**

yes

**Quality:**

3

**Strengths And Weaknesses:**

### **Strengths:**

- **Clear and Well-Structured:** The paper is well-organized, with detailed explanations of the preliminary, intuition, and methodology.

- **Interesting and Novel:** This paper presents a geometrically intuitive solution for multimodal alignment that directly aligns embedding vectors from different modalities by minimizing the area of triangles formed by these vectors. The approach leverages the geometric property that smaller triangle areas correspond to better alignment between embedding vectors, providing an intuitive and interpretable framework for multimodal alignment.

- **Extensive Experiments:** The authors conduct extensive experiments, and TRIANGLE achieves state-of-the-art performance compared to a wide range of baselines, showing significant improvements across multiple datasets. These results demonstrate the effectiveness of the TRIANGLE approach.

### **Weaknesses:**

- **Lack Theoretical Analysis:** While the proposed method is intuitive and demonstrates effectiveness, its theoretical foundation requires further clarification. Specifically, the connection between minimizing triangle area and achieving better alignment lacks sufficient justification—why does minimizing the triangle area lead to better alignment? I acknowledge that a formal proof may be beyond the scope of this paper; however, I suggest the authors provide a comparative analysis of how the TRIANGLE objective affects the cosine similarity of aligned latent representations relative to previous baselines to substantiate the geometric intuition. Additionally, including visualizations of the latent space transformations and empirical analysis of how triangle area reduction correlates with alignment performance would strengthen the theoretical grounding of the submission.

- **Unclear Problem Definition and Annotation:** Some definitions and annotations in Section 3 are introduced without clear explanation or prior clarification. I recommend that the authors refine this section to improve clarity and readability.

- **Insufficient Ablation Study:** While the paper provides an ablation study on key methodological components, investigating the effects of the $L_{DTM}$ loss function and the weight regularization parameter $\alpha$, a more comprehensive analysis would strengthen the submission. Specifically, the authors only examine the significance of the $L_{DTM}$ loss function, but it remains unclear whether assigning different weights $\lambda$ could lead to better performance.

- **Formatting:** Please use` \citep{}` for parenthetical citations instead of `\citet{}` throughout the manuscript.

---

> ### Author Rebuttal · Authors · 2025-07-30
>
> **Theoretical Analysis.** Thanks for giving us the opportunity to better clarify the link between area minimization and better alignment. The intuition is provided in Figure 3 (b) in the paper, where we show that as the area gets smaller, the three vectors collides. Formally, since the area is defined as:
>
> $$    A = \frac{1}{2} \sqrt{\langle \mathbf{u}, \mathbf{u} \rangle \langle \mathbf{v}, \mathbf{v} \rangle - \langle \mathbf{u}, \mathbf{v} \rangle^2},$$
>
> in which $\mathbf{u} = \mathbf{x} - \mathbf{y}$, $\mathbf{v} = \mathbf{x} - \mathbf{z}$, we obtain $A=0$ if $(\mathbf{x} - \mathbf{y})$ and  $(\mathbf{x} - \mathbf{z})$ are linearly dependent, that is when the three embedding vectors coincides having cross-cosine similarities equal to 1, thus representing perfect alignment.
>
> To prove our theoretical claims, we conduct additional experiments to show how minimizing the triangle area brings improvements in the alignment of the latent space. Specifically, we test TRIANGLE in the Touch-Vision-Language Dataset [1], which comprises in-the-wild vision-touch pairs with language labels. The current state-of-the-art model, Touch-Vision-Language (TVL), leverages pairwise cosine similarity among all the pairwise, training the tactile encoder to align it to the vision and text encoder from CLIP. We retrained TVL (first row in the table) and we train the same models with the proposed TRIANGLE loss (second row in the table). As it is clear from the scores, **TRIANGLE improves the alignment of the pairwise comparison too**, better aligning vision and tactile modalities. What is more, by involving our three-modal loss, **we can also unlock a novel task, Vision to Tactile&Text, that allows us to effectively align the three modalities altogether**, building a joint latent space. This empirically demonstrates that minimizing the triangle area also improves the cosine similarity alignment between pairs of modalities.
>
> |          | Acc@1    | Acc@5    | Vision-Tactile Acc@1 | Vision-Tactile Acc@5 | Vision-Tactile&Text Acc@1 | Vision-Tactile&Text Acc@5 |
> |----------|----------|----------|----------------------|----------------------|---------------------------|---------------------------|
> | TVL      | 36.7     | 53.3     | 79.9                 | 93.1                 | -                   | -                   |
> | TRIANGLE | **83.1** | **94.6** | **82.8**             | **94.6**             | **83.5**                  | **94.7**                  |
>
> [1] Letian Fu, et al. "A Touch, Vision, and Language Dataset for Multimodal Alignment", ICML 2024.
>
> **Unclear definitions.** We have revised and updated the notations and the definitions in Section 3, thank you for pointing this out!
>
> **Ablation studies.** According to the Reviewer's suggestion, we perform an additional ablation study on the value of $\lambda$ that regulates the influence of $L_{DTM}$ in the total loss in Eq 8.
> In all the experiments conducted in the paper, we set $\lambda=0.1$, inhereting this value from previous works (VAST and GRAM). In this ablation, we test different values ranging in (0.0, 0.1, 0.3, 0.5, 1.0).
> From the result table below, it is clear that the setting with $\lambda=0.1$ obtains the best results in 3 over 4 scores, thus confirming the setting choice of the experiments in the main paper.
>
> | Training from scratch on MSRVTT | $\lambda$         | T2AV R@1 | T2AV R@10 | AV2T R@1 | AV2T R@10 |
> |-----------------|---------------|----------|-----------|----------|-----------|
> | TRIANGLE |   0.0    |  33.3     | 74.4      | 40.4     | **81.7**  |
> | TRIANGLE |   0.1    | **39.4** | **81.8**  | **41.9** | 80.0 |
> | TRIANGLE |   0.3    |  20.9   | 71.6     | 20.7     |   72.4    |
> | TRIANGLE |   0.5    |  24.0   |  74.2    |   22.2   |  71.9     |
> | TRIANGLE |   1.0    |  38.3   |   81.2   |  36.9    |  78.4     |
>
>
> We added this ablation in the revised manuscript, thanks again for the suggestion!
>
> **Formatting** Thanks for pointing this out! We have updated the citation formatting according to Reviewer's suggestion.

---

> > ### Comment · Reviewer_tnky · 2025-08-05
> >
> > Thank you for your response. The rebuttal has largely addressed my concerns. However, I still recommend that the authors include an illustration to clarify the relationship between cosine similarity, the TRIANGLE objective, and model performance.

---

> > > ### Author Response · Authors · 2025-08-05
> > >
> > > We thank the Reviewer for the constructive and valuable feedback. As suggested, we have included an illustration in the final version of the manuscript to clarify the relationship between cosine similarity, the TRIANGLE objective, and model performance. Unfortunately, due to NeurIPS policy on external links during the rebuttal phase, we are unable to share this figure at this time, but it will be available in the camera-ready version.
> > >
> > > We are pleased to hear that we have largely addressed the Reviewer’s concerns regarding the theoretical analysis. Together with the additional ablation study requested, we believe that the Reviewer’s comments has helped strengthen the overall contribution of our work.
> > >
> > > If there are no further concerns or questions, we kindly ask the Reviewer to consider revising the assigned score.

---

> > > > ### Comment · Reviewer_tnky · 2025-08-06
> > > >
> > > > Based on your rebuttal, I decided to update my rating to 5. Good luck.

---

### Official Review · Reviewer_mfjz · 2025-07-02

**Clarity:** 3
**Significance:** 3
**Originality:** 2
**Rating:** 5
**Confidence:** 3

**Summary:**

The paper introduces a novel method for multimodal learning, named TRIANGLE, which is based on minimizing the area of the triangle formed by the embeddings of three modalities in an n-dimensional space. Unlike traditional approaches that use pairwise cosine similarity and rely on anchor-based alignment, TRIANGLE aligns all three modalities jointly. The triangle area provides a geometrically meaningful indicator of alignment: a smaller area corresponds to closer embeddings and hence better alignment, while a larger area suggests that at least one modality is misaligned. The authors evaluate their method extensively under three different evaluation settings: vanilla experiment, zero-shot retrieval and classification, and training from scratch. The empirical studies are conducted on the alignment problem between video, audio, and text modalities, demonstrating improvements over state-of-the-art methods. The experiments show superior performance across three different setups, with a negligible increase in computational time.

**Questions:**

1) Would it be possible to test the method on a different set of modalities to support the general claim?
2) Can you improve the discussion on why TRIANGLE outperforms n-model losses methods?
3) How do you handle the near-degeneracy triangle case that may be caused by collinearity? Does the method remain stable in such a scenario?

**Ethical Concerns:**

["NO or VERY MINOR ethics concerns only"]

**Final Justification:**

I raise my score to accept. The questions from the reviewers were properly addressed, and the additional experiments are good. I feel like the interpretability part can be improved, but it's not a big issue.

**Limitations:**

The method is limited by the number of modalities that can be used. However, a solution is proposed in the Limitation section in the appendix.

**Quality:**

3

**Strengths And Weaknesses:**

Strengths: The authors propose a novel method to better align three heterogeneous modalities. To overcome the limitations imposed by the cosine similarity, they improve the contrastive learning loss proposed in the state of the art by using the triangle area as a similarity metric, which allows the consideration of three modalities. The mathematical formulation, along with the proofs and possible regularizations, is discussed throughout the paper and in the appendix. The method has been extensively tested on many datasets and three different setups, namely: “vanilla experiment” on a controlled environment (MNIST dataset and variants), “zero-shot retrieval” exploiting pretraining, and “learning from scratch”. In all the proposed experiments, the method shows a significant improvement with respect to the state of the art or, in the worst case, a comparable result (only considering R@1 in the third experiment). Moreover, the method shows a faster convergence in the vanilla experiment with respect to the state of the art.

Weaknesses: The limitations and future work should be discussed in the main paper. Even if the method has been extensively evaluated on many datasets, they always present the same modalities (text, video, and audio), would it be possible to test the method on a different set of modalites to show its performance is not only applicable on that limited set of modalities (In the introduction TRIANGLE is proposed as a generic three modalities method)?  In the paper, the author claims an improvement on the interpretability side. Knowing the importance of that word for the machine learning community, I expect it to be discussed a bit more throughout the paper. To conclude, while TRIANGLE outperforms GRAM and Symile in three-modal tasks, a more in-depth discussion or theoretical comparison on why a tailored triplet objective is inherently superior to general n-modal losses (beyond empirical results) would strengthen the paper.

---

> ### Author Rebuttal · Authors · 2025-07-30
>
> **Experiments with different modalities.** We test TRIANGLE on a different set of modalities in the Touch-Vision-Language Dataset [1], which comprises in-the-wild vision-touch pairs with language labels. The current state-of-the-art model, Touch-Vision-Language (TVL), leverages pairwise cosine similarity among all the pairwise, training the tactile encoder to align it to the vision and text encoder from CLIP. We retrained TVL (first row in the table) and we train the same models with the proposed TRIANGLE loss (second row in the table). As it is clear from the scores, **TRIANGLE outperforms TVL in the standard two-modal task**, better aligning vision and tactile modalities. What is more, by involving our three-modal loss, **we can also unlock a novel task, Vision to Tactile&Text, that allows us to effectively align the three modalities altogether**, building a joint latent space.
>
> |          | Acc@1    | Acc@5    | Vision-Tactile Acc@1 | Vision-Tactile Acc@5 | Vision-Tactile&Text Acc@1 | Vision-Tactile&Text Acc@5 |
> |----------|----------|----------|----------------------|----------------------|---------------------------|---------------------------|
> | TVL      | 36.7     | 53.3     | 79.9                 | 93.1                 | -                   | -                   |
> | TRIANGLE | **83.1** | **94.6** | **82.8**             | **94.6**             | **83.5**                  | **94.7**                  |
>
> [1] Letian Fu, et al. "A Touch, Vision, and Language Dataset for Multimodal Alignment", ICML 2024.
>
> **On the interpretability** We appreciate the suggestion for a deeper discussion on the interpretability. Conventional methods that fuses modalities through MLP layers (like VAST) or anchor-based pairwise cosines (like LanguageBind, ImageBind) produces scores whose internal meaning either depends on learned parameters or on different measures of alignment (i.e., for the three modal case, they need to compute the pairwise cosine between each of the modalities, resulting in three different measures, M1-M2, M1-M3, M2-M3), without having a direct and clear information on the **overall alignment of the space**. In our work, instead, we refer to "interpretability” as **having a direct, human-readable geometric indicator of three-way alignment**, which is the area of the triangle formed by the modality embeddings. Indeed, intuitively, a small area implies that all three modalities point to similar semantics, while a large area signals disagreement among at least two of them. In this sense, the TRIANGLE measure of alignment is more interpretable than fusing strategies or multiple pairwise cosine similarities. We would like to thank the Reviewer once again, we include this discussion in the revised version of the paper.
>
> **On the triplet objective.** We would like to thank the Reviewer for giving us the opportunity to further discuss this important aspect of our work.
>
> Let us assume that we have three modalities, namely text, audio, and vision, although the same analysis can be applied to any other set of modalities.
>
> All the three methods (i.e., Symile, GRAM, and TRIANGLE) involve a single unimodal encoder for each modality, and the normalized extracted embeddings are $\textbf{t}$ text, $\textbf{v}$ vision, and $\textbf{a}$ audio. We report only $L_{D2T}$; the same holds for $L_{T2D}$. In what follows, $\langle \cdot, \cdot \rangle$ denotes the dot product between two vectors of the same dimensionality.
>
> **Symile** formulates the loss function to ensure that all its components contribute to maximizing a lower bound on the total correlation among all modalities involved in the training. It relies on the multilinear inner product (MIP), which is not a measure of the geometric representations similarity, as stated in their paper. We found that, in real-world scenarios, Symile performance is weak, likely due to the lack of geometric principles behind their method. Formally, Symile loss can be defined as:
>
> $$L_{D2T\-Symile} =-\frac{1}{B}\sum_{i=1}^{B}\log\frac{\exp(\langle \textbf{t}_i, \textbf{v}_i, \textbf{a}_i \rangle /\tau)}{\sum^K_j\exp(\langle \textbf{t}_j, \textbf{v}_i, \textbf{a}_i \rangle/\tau)}$$
>
> Where $\langle \textbf{t}_i, \textbf{v}_i, \textbf{a}_i \rangle$ is the MIP, which is an extension of the classic dot product and is defined as $\langle \textbf{t}_i, \textbf{v}_i, \textbf{a}_i \rangle = \sum^D_d{\textbf{t}_i}_d \cdot {\textbf{v}_i}_d \cdot {\textbf{a}_i}_d$, with $D$ is the dimensionality of embeddings.
>
> **GRAM** introduces a geometrical approach for the multimodal pretraining of encoders based on the volume of the k-dimensional parallelotope spanned by the embeddings of the different modalities as
>
> $$L_{D2T\-GRAM}=-\frac{1}{B}\sum_{i=1}^{B}\log\frac{\exp(-Vol(\textbf{t}_i, \textbf{v}_i, \textbf{a}_i)/\tau)}{\sum^K_j\exp(-Vol(\textbf{t}_j, \textbf{v}_i, \textbf{a}_i)/\tau)}$$
>
> Where $Vol(\textbf{t}_i,\textbf{v}_i,\textbf{a}_i)= \sqrt{\det(\mathbf{G})}$; $\mathbf{G}=\mathbf{Z}^{\top}\mathbf{Z}$; $\mathbf{Z}=(\mathbf{t}_i,\mathbf{v}_i,\mathbf{a}_i)$. We will discuss the details of this method few lines below.
>
> **TRIANGLE** also is rooted on geometrical concepts, however, its formulation is different from GRAM. The area of the triangle built by the embeddings in the latent space is taken into consideration as an alignment metric of the embeddings themself, resulting in:
>
> $$L_{D2T\-TRIANGLE}=-\frac{1}{B}\sum_{i=1}^{B}\log\frac{\exp(-A(\textbf{t}_i, \textbf{v}_i, \textbf{a}_i)/\tau)}{\sum^K_j\exp(-A(\textbf{t}_j, \textbf{v}_i, \textbf{a}_i)/\tau)}$$
>
> here $A(\textbf{t}_i,\textbf{v}_i,\textbf{a}_i) = \frac{1}{2}\sqrt{\langle \mathbf{u}, \mathbf{u} \rangle \langle \mathbf{v}, \mathbf{v} \rangle - \langle \mathbf{u}, \mathbf{v} \rangle^2}$, in which  $\mathbf{u} = \mathbf{t}_i - \mathbf{a_i}$, $\mathbf{v} = \mathbf{t}_i - \mathbf{v}_i$ are two triangle sides computed among the three embeddings $\mathbf{t}_i, \mathbf{v}_i,$ and $\mathbf{a}_i$ of the three modalities.
> Although TRIANGLE can be mistakenly seen as a special case of GRAM when $n=3$, it can be shown that the two objective functions are fundamentally different mathematically and operate on distinct aspects of the geometrical alignment.
>
> Going into details, by expanding the volume formula and computing all the multiplication needed to compute the determinant of the matrix $\mathbf{G}$, it can be seen that
>
> $$Vol(\textbf{t}_i,\textbf{v}_i,\textbf{a}_i)= \sqrt{1 - \textbf{v}_i\textbf{a}_i^2 - \textbf{t}_i\textbf{a}_i^2 - \textbf{t}_i\textbf{v}_i^2 + 2 \cdot \textbf{t}_i\textbf{a}_i \cdot \textbf{a}_i\textbf{v}_i \cdot \textbf{v}_i\textbf{t}_i}$$
>
> Whereas the area of the triangle is:
>
> $$A(\textbf{t}_i,\textbf{v}_i,\textbf{a}_i) = \frac{1}{2}\sqrt{\langle (\mathbf{t}_i - \mathbf{a_i}), (\mathbf{t}_i - \mathbf{a_i})\rangle \langle (\mathbf{t}_i - \mathbf{v}_i), (\mathbf{t}_i - \mathbf{v}_i) \rangle - \langle (\mathbf{t}_i - \mathbf{a_i}), (\mathbf{t}_i - \mathbf{v}_i) \rangle^2}$$
>
> which is significantly different.
>
> Geometrically, the GRAM equation ($Vol$) is the volume of the parallelotope spanned by the three vectors with one vertex at the origin, measuring how mutually linearly independent the three origin-anchored vectors are. On the contrary, the TRIANGLE area ($A$) formula measures how far apart the  difference vectors are, i.e., the edges of the triangle with vertices at $\mathbf{t}, \mathbf{a}, \mathbf{v}$, not comprising the origin. Therefore, the two losses optimize different geometrical entities and rely on diverse formulations.
> These highlighted differences, when incorporated into contrastive learning loss functions, influence different aspects of the vector alignment, resulting in different downstream performance, favoring the three-modal-specialized TRIANGLE in all the comparisons.
> We have updated the revised paper including this discussion, following the valuable suggestion of the Reviewer.
>
> **On the limitations in main paper** We can definitely move the limitations in the main body and organize the last page accordingly. Thanks for the suggestion.

---

> > ### Comment · Reviewer_mfjz · 2025-08-01
> > **On the collinearity question**
> >
> > Thank you very much for your reply. Could you please reply to question 3 (stability upon collinearity)?

---

> > > ### Author Response · Authors · 2025-08-01
> > > **On the collinearity question**
> > >
> > > We thank the Reviewer for the quick response!
> > > The degeneracy scenario in which all three embeddings lie on the same line only happens in two cases: 1) If all three vectors are identical. In this case, the objective is perfectly minimized, and the overall goal is achieved. 2) If two vectors are identical, and the third lies on the opposite side of the hypersphere (i.e. $\cos(\mathbf{x}, \mathbf{y}) = 1, \cos(\mathbf{x}, \mathbf{z}) = \cos(\mathbf{y},\mathbf{z})=−1$). In this situation, the area is zero too.
> > > However, during training, we have never observed the latter specific situation, as it is extremely rare and specific. The models trained with the proposed loss have not shown ay instabilities and smoothly converge, as we show in the TRIANGLE loss in Fig.5, in the area value during training in Fig.6 of Appendix, and in the performance progression across epochs in Figure 4. In the final version of the paper, we will also include the loss landscape for both the pretraining phase and training from scratch on MSRVTT to strengthen our claims and further prove the stability of the proposed loss.
> > > However, to handle any specific case in downstream tasks (in which the model has to well-behave in the wild), we have indìtroduced a cosine similarity regularization, as described in Section 3.3 of the paper (Downstream Tasks Regularization). By progressively excluding the potential effects of collinearity and placing a little bit more emphasis on pure cosine similarity between the two most affected modalities, the system is able to slightly improve performance, as shown in the ablation studies (Figure 7).

---

### Official Review · Reviewer_u6BX · 2025-07-02

**Clarity:** 3
**Significance:** 3
**Originality:** 3
**Rating:** 5
**Confidence:** 3

**Summary:**

This paper proposes TRIANGLE, a novel method for aligning three modalities (e.g., video, audio, text) in multimodal embedding spaces. The authors observe that existing contrastive approaches rely on pairwise cosine similarity with an anchor modality, lacking direct measures of alignment among all modalities. TRIANGLE introduces a **triangle area-based similarity measure** computed in the higher-dimensional embedding space formed by three modality vectors. This geometric area intuitively represents overall alignment (smaller area indicates better alignment) and is used to define a new contrastive loss. TRIANGLE avoids additional fusion layers and anchors, yielding more interpretable alignment rationales. Extensive evaluations on video-text retrieval, audio-text retrieval, and audio-video classification  demonstrate state-of-the-art performance and robust generalization across downstream tasks.

**Questions:**

You mention potential extensions via convex hull decomposition or sub-triangle summation. Could you provide preliminary results or formal insights on computational complexity and numerical stability in such extensions?

Could you include or release examples of retrieval outputs (queries with top-5 retrieved videos, audios, and captions) to illustrate practical improvements and highlight specific cases where TRIANGLE succeeds over VAST or GRAM?

**Ethical Concerns:**

["NO or VERY MINOR ethics concerns only"]

**Final Justification:**

I found the majority of the rebuttal are well-justified. All my concerns have been addressed. Thus I raise my score to accept.

**Limitations:**

Yes

**Quality:**

3

**Strengths And Weaknesses:**

Strengths
Rigorous formulation with a novel, theoretically grounded geometric similarity measure.

Extensive experimental validation across seven datasets and multiple tasks, demonstrating consistent improvements.
Statistical significance testing is included, strengthening empirical claims.
Well-structured explanation with intuitive figures illustrating the triangle area concept.

Weaknesses
Results on training from scratch are promising but still slightly below pretraining-based baselines (e.g. Table 3 MSR-VTT T2AV task).

Figures could include qualitative retrieval examples (e.g., video-audio-text samples with retrieved captions) to improve interpretability for practical deployment.

Minor typos and formatting inconsistencies in equations

---

> ### Author Rebuttal · Authors · 2025-07-30
>
> **On the training from scratch.** Table 3 presents the results of training all the models only on the MSRVTT dataset from scratch (that is, by randomly initializing all model weights), then evaluating on the test set. The purpose of this table is to provide a direct comparison with other methods, isolating the effect of loss functions and removing the influence of other factors such as pretraining data and specific training pipelines, not to achieve the best results overall. In this controlled setting, we demonstrate the superiority of TRIANGLE compared to common multimodal contrastive loss functions.
>
> In Table 1, the models are pretrained using large datasets (e.g., VAST on 27M of videos, LanguageBind on 10M, and so on), while the training set of MSRVTT used in Table 3 contains only 9k videos. Therefore, the performance gap between Table 1 and 3 is given by the huge difference in dataset size, as it is well-known that pretraining data plays a crucial role in achieving state-of-the-art performance.
>
> Overall, TRIANGLE obtains consistent results even when the training dataset is smaller, proving the effectiveness of the proposed method once again.
>
> **On the extension via convex hull.** We conduct preliminary results on the AV-MNIST dataset adding a fourth modality, that is the Colored MNIST, so to have a total set of 4 modalities (greyscale MNIST images, colored MNIST images, audio MNIST spectrograms, textual labels), processed with four different encoders (details on the encoders in the main paper). We compute the convex-hull-based area loss using fan-triangulation from a fixed anchor (text), avoiding explicit convex hull construction. For $K$ modalities (4 in our case), this requires computing $(K−2)$ triangle areas (2 computations in our case), each with $O(D)$ complexity (where $D$ is the embedding dimension, 3 in our case). Therefore, the per-sample complexity is $O((K−2)·D)$, and the full contrastive matrix costs $O(B^2·(K−2)·D)$, with $B$ batch size, where the TRIANGLE complexity in the case single case of $K=3$ is equal to $O(B^2D)$. We observe no instabilities in this experiment. However, being a preliminary experiment, this solution is still an approximation and should be further developed and tested to handle all the special cases, but we achieved R@1 56.5 and R@5 97.5.
>
> We include this discussion in the revised paper and we plan to further explore it in future works, thanks for raising this interesting question!
>
>
> **Examples of retrieval outputs.** Thanks to the Reviewer for the valuable suggestion. We agree that incorporating a concrete, real-world example would strengthen the contribution of our work. In response, we conducted an additional inspection on the VGGSound dataset. This dataset consists of multimodal videos, each comprising image frames and corresponding audio samples, with an associated textual class label. VGGSound includes a total of 309 distinct classes and we already report R@1 and R@10 in Tab.2.
>
> Here, we introduce an additional metrics, the Mean Retrieval Rank (MRR), which reflects the average rank position of the ground-truth class among all retrieved classes across the entire test set. The results are as follows:
>
> | Method    | Mean Retrieval Rank (↓) |
> |-----------|--------------------------|
> | VAST      | 27.3                    |
> | GRAM      | 16.7                    |
> | TRIANGLE | **14.8**                 |
>
> These results demonstrate that TRIANGLE consistently ranks the relevant content more effectively compared to other models.
>
> In addition, we also manually inspected several retrieved examples. Due to NeurIPS rebuttal policy restrictions, we are currently unable to include links to external multimodal content or images. However, we will include visual examples in the final version of the paper to further illustrate the model performance, and report here only the textual outcomes.
>
> We briefly describe the outcomes from three representative multimodal examples:
>
> **SAMPLE 1**
> VGGSOUND ID:   $\text{--U7joUcTCo}$ from T=0s (i.e., from second 0 of the video)  Ground truth class: **'People coughing'**.
>
> 	Results for **TRIANGLE**
>     Rank of ground truth class: **3rd**.
>     TOP 5 Retrieved classes: ['people giggling', 'people belly laughing', **'people coughing'**, 'baby laughter', 'people sobbing']
>
> 	Results for **VAST**
>     Rank of ground truth class: **22nd**
>     TOP 5 Retrieved classes: ['people belly laughing', 'people giggling', 'people whistling', 'male speech man speaking', 'people cheering']
>
>
> **SAMPLE 2**
> VGGSOUND ID:  $\text{-0vPFx-wRRI}$ from T=30s Ground truth class: **'people finger snapping'**
>
> 	Results for **TRIANGLE**
>     Rank of ground truth class: **1st**
>     TOP 5 Retrieved classes: [**'people finger snapping'**, 'people eating apple', 'male singing', 'playing castanets', 'people eating crisps']
>
>
> 	Results for **VAST**
>     Rank of ground truth class: **22nd**
>     TOP 5 Retrieved classes: ['male singing', 'child speech kid speaking', 'child singing', 'playing castanets', 'people giggling']
>
>
>
> **SAMPLE 3**
> VGGSOUND ID:  $\text{2Yo0MGR2kmA}$ from T=31s Ground truth class:  **'Playing bagpipes'**
>
> 	Results for **TRIANGLE**
>     Rank of ground truth class: **1st**
>     TOP 5 Retrieved classes: [**'playing bagpipes'**, 'playing ukulele', 'playing banjo', 'playing sitar', 'playing steelpan']
>
>
> 	Results for **VAST**
>     Rank of ground truth class: **30th**
>     TOP 5 Retrieved classes: ['playing ukulele', 'playing sitar', 'playing mandolin', 'playing steel guitar slide guitar']
>
>
> From our qualitative analysis, we consistently observe that TRIANGLE effectively integrates information from all three modalities (visual, audio, and text). This results in superior performance on tasks such as video classification and reasoning, where a holistic understanding across modalities is crucial.
>
> A particularly illustrative example is **Sample 3** from our previous analysis. In the video, a man holding a guitar or ukulele appears in the foreground, while another man with a bagpipe is seen in the background. However, upon listening to the audio, it becomes clear that only the bagpipe is being played. TRIANGLE successfully captures this multimodal context and correctly classifies the video as “Playing bagpipes”, whereas VAST misclassifies it as “Playing ukulele” by likely over-relying on visual cues.
>
> **Typos and formatting.** We double-checked the whole paper and revised typos and formatting errors, thank you!

---

> ### Comment · Area_Chair_o4zC · 2025-08-05
> **Rebuttal**
>
> Dear Reviewer,
>
> Authors have responded to your review. Please let the authors and us know if your questions and concerns have been addressed satisfactorily. If there are any outstanding issues that haven't yet been addressed, please highlight them. And if you have any further questions, comments, observations or suggestions to the authors based on the rebuttal, please post them now.
>
> -- AC

---

> > ### Author Response · Authors · 2025-08-06
> >
> > Thank you again for your time and for your insightful and thorough review of our work.
> > As the discussion period deadline is approaching, we wanted to respectfully follow up on our detailed rebuttal, where we sought to address every concern you raised.
> >
> > To summarize, in our response we have:
> >
> > - Clarified the role of the training-from-scratch experiments, explaining that the purpose is to isolate the effect of loss functions without relying on large-scale pretraining, and demonstrating that TRIANGLE outperforms other methods even under this constrained setting.
> >
> > - Provided preliminary insights into the extension via convex hull decomposition.
> >
> > - Introduced new qualitative and quantitative retrieval examples on the VGGSound dataset.
> >
> > - Reviewed and corrected formatting inconsistencies and minor typos throughout the manuscript for improved clarity and presentation.
> >
> > We believe these detailed responses and new results further reinforce the validity and usefulness of our proposed TRIANGLE method. We are grateful for your feedback and happy to answer any additional questions you may have.
> >
> > Thank you again for your time and consideration.

---

### Decision · Program_Chairs · 2025-09-17

**Decision:**

Accept (poster)

**Comment:**

This work addresses the problem of aligning multiple (>= 3) modalities in multimodal learning. It argues that prior work in multimodal learning have severe limitations as they rely on cosine similarity between modalities and select an anchor modality to which other modalities are aligned one-by-one. It claims that due to such an approach, such methods might not sufficiently well align non-anchor modalities. Along the lines of recent works (GRAM, Symile), the current work proposes an alternative similarity metric for multimodal learning that scales to arbitrary number of modalities. The key idea is to consider the area of the triangle formed by the embeddings of the three modalities as a measure of their dissimilarity and align the modalities jointly by minimizing the area. It is argued that the area of the triangle provides a geometrically meaningful indicator of alignment with smaller areas indicating better alignment. As area can be small also when the modalities are not closely aligned (e.g. flat triangles), the work employs cosine regularization of the two most important modalities for the downstream task. The loss function has two components - first is a contrastive loss based on triangle area and the second is a data text loss. The proposed method is evaluated in under different evaluation settings - vanilla, zero-shot retrieval and classification, and training from scratch. Experiments consider text, video and audio as the three modalities for alignment. In the rebuttal phase, a fourth modality, touch, was also included. Results of the experimental evaluation are very promising and demonstrates significant empirical gains over several baselines including GRAM and Symile.


Strengths:

Novelty

mfjz: "authors propose a novel method to better align three heterogeneous modalities"

u6BX: a novel, theoretically grounded geometric similarity measure

Tnky: "presents a geometrically intuitive solution for multimodal alignment"

bcfQ : "TRIANGLE is intuitive, leveraging basic geometric concepts like area"

Encouraging results

u6BX: "Extensive experimental validation across seven datasets and multiple tasks, demonstrating consistent improvements."

mfjz: "The method has been extensively tested on many datasets and three different setups.. the method shows a significant improvement with respect to the state of the art or, in the worst case, a comparable result"

mfjz: "the method shows a faster convergence in the vanilla experiment with respect to the state of the art"

Tnky: "achieves state-of-the-art performance compared to a wide range of baselines, showing significant improvements across multiple datasets."

bcfQ:  "for the tasks on which it is evaluated, TRIANGLE demonstrates strong performance"


Weaknesses:

Concern about results for training from scratch

u6BX: "slightly below pretraining-based baselines"

Concern about complexity and stability of extension to K > 3 modalities.

Concern about effectiveness to modalities other than text, audio and video

mfjz: "test the method on a different set of modalites to show its performance is not only applicable on that limited set of modalities"

Concern about some claims made but unsubstantiated

mfjz: "author claims an improvement on the interpretability side. Knowing the importance of that word for the machine learning community, I expect it to be discussed a bit more throughout the paper"

bcfQ: "The paper claims that “this [anchor-based] alignment only ensures that all modalities are aligned with the anchor, without providing any indicators for the alignment between non-anchor modalities”. To this reviewer’s understanding, this is not true."


Concerns about ablation

Tnky: "investigating the effects of the  loss function and the weight regularization parameter , a more comprehensive analysis would strengthen the submission."

Lack of theoretical analysis of the proposed approach

 mfjz: "a more in-depth discussion or theoretical comparison on why a tailored triplet objective is inherently superior to general n-modal losses"

Tnky: "the connection between minimizing triangle area and achieving better alignment lacks sufficient justification"

Tnky: "provide a comparative analysis of how the TRIANGLE objective affects the cosine similarity of aligned latent representations relative to previous baselines"

Concerns about the soundness of the main idea

mfjz: "How do you handle the near-degeneracy triangle case that may be caused by collinearity? Does the method remain stable in such a scenario?"

Lack of qualitative analysis

u6BX: "extensions via convex hull decomposition or sub-triangle summation. Could you provide preliminary results or formal insights on computational complexity and numerical stability in such extensions?"


Final Comments:
This work addresses an important problem and provides a principled solution that is simple to implement and doesn't introduce significant computational overheads. The main strength of the work is its empirical gains over prior work on the datasets used in the experimental study. Claims on interpretability are not well-substantiated, theoretical understanding of how the proposed area-based loss is superior to prior art is lacking, and qualitative analysis of the results is missing. However, reviewers are positive about the novelty, methodology and experimental evaluation. During the rebuttal phase, authors and reviewers had a productive discussion resolving most of the concerns expressed in the reviews.